# Large-scale determination of previously unsolved protein structures using evolutionary information

Sergey Ovchinnikov[1], Lisa Kinch[2], Hahnbeom Park[1], Yuxing Liao[3], Jimin Pei[2], David E Kim[1], Hetunandan Kamisetty[4], Nick V Grishin[2,3], David Baker[1,5]*

[1]Department of Biochemistry, University of Washington, Seattle, United States; [2]Howard Hughes Medical Institute, University of Texas Southwestern Medical Center, Dallas, United States; [3]Department of Biophysics, Department of Biochemistry, University of Texas Southwestern Medical Center, Dallas, United States; [4]Facebook Inc., Seattle, United States; [5]Howard Hughes Medical Institute, University of Washington, Seattle, United States

**Abstract** The prediction of the structures of proteins without detectable sequence similarity to any protein of known structure remains an outstanding scientific challenge. Here we report significant progress in this area. We first describe de novo blind structure predictions of unprecedented accuracy we made for two proteins in large families in the recent CASP11 blind test of protein structure prediction methods by incorporating residue–residue co-evolution information in the Rosetta structure prediction program. We then describe the use of this method to generate structure models for 58 of the 121 large protein families in prokaryotes for which three-dimensional structures are not available. These models, which are posted online for public access, provide structural information for the over 400,000 proteins belonging to the 58 families and suggest hypotheses about mechanism for the subset for which the function is known, and hypotheses about function for the remainder.

*For correspondence: dabaker@uw.edu

Competing interests: The authors declare that no competing interests exist.

## Introduction

Despite substantial efforts over decades, high-resolution structure prediction is currently limited to proteins that have homologs of known structure, or small proteins where thorough sampling of the conformational space is possible (<100 residues; even in this case, predictions can be very inaccurate). For roughly 41% of protein families, there is currently no member with known structure (*Kamisetty et al., 2013*). While high-resolution ab initio structure prediction has remained a challenge, considerable success has been achieved in generating high-accuracy models when sparse experimental data are available to constrain the space of conformations to be sampled. This additional information, in combination with a reasonably accurate energy function, has enabled the determination of high-resolution structures for much larger proteins (*Raman et al., 2009*; *DiMaio et al., 2011*; *Lange et al., 2012*).

Recent work has shown that residue–residue contacts can be accurately inferred from co-evolution patterns in sequences of related proteins (*Marks et al., 2011*; *Morcos et al., 2011*; *Hopf et al., 2012*; *Jones et al., 2012*; *Marks et al., 2012*; *Nugent and Jones, 2012*; *Sułkowska et al., 2012*; *Kamisetty et al., 2013*). While early approaches estimated these restraints by inverting a covariance matrix (*Marks et al., 2011*; *Morcos et al., 2011*; *Jones et al., 2012*), subsequent studies have shown that a pseudo-likelihood (PLM)-based approach (*Balakrishnan et al., 2011*) results in more accurate predictions (*Ekeberg et al., 2013*; *Kamisetty et al., 2013*). Distance restraints derived from such

**eLife digest** Proteins are long chains made up of small building blocks called amino acids. These chains fold up in various ways to form the three-dimensional structures that proteins need to be able work properly. Therefore, to understand how a protein works it is important to determine its structure, but this is very challenging. It is possible to predict the structure of a protein with high accuracy if previous experiments have revealed the structure of a similar protein. However, for almost half of all known families of proteins, there are currently no members whose structures have been solved.

The three-dimensional shape of a protein is determined by interactions between various amino acids. During evolution, the structure and activity of proteins often remain the same across species, even if the amino acid sequences have changed. This is because pairs of amino acids that interact with each other tend to 'co-evolve'; that is, if one amino acid changes, then the second amino acid also changes in order to accommodate it. By identifying these pairs of co-evolving amino acids, it is possible to work out which amino acids are close to each other in the three-dimensional structure of the protein. This information can be used to generate a structural model of a protein using computational methods.

Now, Ovchinnikov et al. developed a new method to predict the structures of proteins that combines data on the co-evolution of amino acids, as identified by GREMLIN with the structural prediction software called Rosetta. A community-wide experiment called CASP—which tests different methods of protein prediction—showed that, in two cases, this new method works much better than anything previously used to predict the structures of proteins. Ovchinnikov et al. then used this method to make models for proteins belonging to 58 different protein families with currently unknown structures.

These predictions were found to be highly accurate and the protein families each have thousands of members, so Ovchinnikov et al.'s findings are expected to be useful to researchers in a wide variety of research areas. A future challenge is to extend these methods to the many protein families that have hundreds rather than thousands of members.

predictions have been used to model a wide range of unknown protein structures (*Hayat et al., 2014*; *Wickles et al., 2014*; *Abriata, 2015*; *Antala et al., 2015*; *Hopf et al., 2015*; *Tian et al., 2015*) and protein–protein complexes (*Ovchinnikov et al., 2014*; *Hopf et al., 2014*). However, while the generated structures often recapitulate the fold of the target protein, it has not been clear whether such methods can yield high-accuracy models of complex protein structures.

## Results

### CASP11 predictions

In the recent CASP11 (11th critical assessment of techniques for protein structure prediction) blind test of protein structure prediction methods, we predicted the structures of proteins from large families with no representatives of known structure by integrating co-evolution derived contact information from GREMLIN (*Kamisetty et al., 2013*) into the Rosetta structure prediction methodology (*Simons et al., 1999*; *Rohl et al., 2004*; *Raman et al., 2009*). Starting from an extended polypeptide chain, Monte Carlo + Minimization searches through conformations with local structure consistent with the local sequence were carried out, optimizing first a low-resolution energy function favoring hydrophobic burial and backbone hydrogen bonding, and second a detailed all atom energy function describing hydrogen bonding and electrostatic interactions, van der Waals interactions, and solvation (*Das and Baker, 2008*). In the first phase, sampling was carried out in internal coordinates (the backbone torsion angles), and hence, to avoid loss of sampling efficiency by early formation of contacts between residues distant along the sequence, predicted contact information was first added for residues close along the chain and subsequently for residues with increasing sequence separation. The contact information was implemented through residue–residue distance restraints whose strength and shape were functions of the strength of the evolutionary covariance between the residues (see 'Materials and methods'). Large numbers of independent

trajectories were carried out using the Rosetta@Home distributed computing project, and the lowest energy (Rosetta all atom energy plus evolutionary restraint fit) models were recombined and further optimized using a new iterative version (see 'Materials and methods') of the RosettaCM hybridization protocol (*Song et al., 2013*; *Kim et al., 2014*). The five lowest energy structures were submitted as predictions to the CASP organizers.

When several months later the actual structures of these proteins were revealed, the predictions were found to be considerably more accurate than any previous predictions made in the 20 years of CASP experiments for proteins over 100 amino acids that lack homologs of known structure. Two particularly striking examples are shown in *Figure 1*; the prediction for the complex 256 residue structure of T0806 is 3.6 Cα-RMSD from the crystal structure (2.9 Å over 223 residues), and the prediction for the 108 residue T0824 is 4.2 Cα-RMSD from the crystal structure (2.7 Å over 77 residues). The models accurately recapitulate the complex topologies of the proteins. Due to time restraints, the calculations could not be run to convergence during CASP; with additional sampling, the lowest energy model for T0806 has an RMSD of 2.1 Å over 245 residues to the experimentally determined structure. Both the co-evolution derived contacts and the new iterative hybridization protocol were critical to obtaining higher accuracy models: Rosetta calculations without constraints failed to converge (data not shown), and the ROBETTA server models generated without the hybridization step were considerably less accurate (11.6 vs 3.6 Cα-RMSD for T0806 and 14.0 vs 4.2 Cα-RMSD for T0824).

## Prediction of structures for large protein families

Having found that protein structures can accurately be modeled using co-evolution information, we set out to build models for representatives of large protein families in bacteria with no detectable structural homologs. To facilitate evaluation of such models, we developed a length-independent measure of the fit between a set of predicted contacts and a model: the ratio of the total GREMLIN score of the model to the score expected if it were the native structure (Rc, see 'Materials and methods').

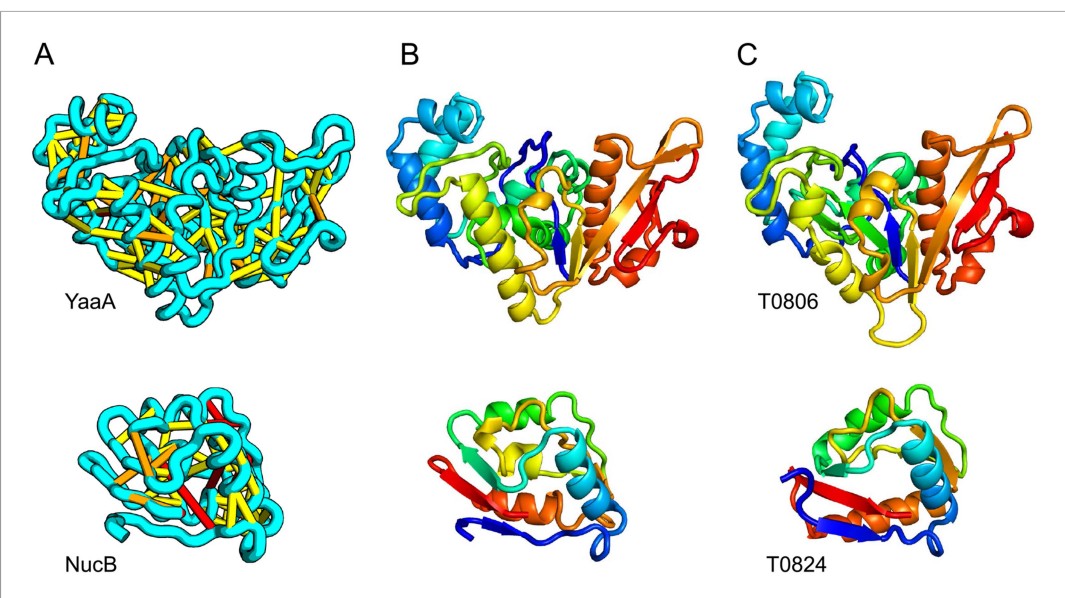

**Figure 1**. Accurate blind structure prediction of CASP11 targets T0806 and T0824. (**A**) Location of the most strongly co-evolving residue pairs. Lines connect residue pairs with normalized coupling strength greater than 1.0; yellow, distance less than 5 Å; orange, distance less than 10 Å and red, greater than 10 Å in the models. (**B**) CASP11 submitted models, colored from N to C terminus (blue to red). (**C**) X-ray crystal structures. For T0806, the Cα RMSD over the full-length protein is 3.6 Å and 2.9 Å over 223 aligned residues. For T0824; the Cα RMSD over the full-length protein is 4.2 Å and 2.7 Å over 77 aligned residues. For statistics on all five models submitted during CASP, see *Figure 1—source data 1*.

The following source data is available for figure 1:

**Source data 1**. The Cα-RMSD and GDT-TS calculations are over the full-length sequence.

We chose to focus on families with at least 4× (protein length) sequences to ensure that the predicted contacts have high accuracy (*Kamisetty et al., 2013*; at 4L sequences, the top 1.5L contacts are on average 50% correct). Families with detectable structure homologs were excluded using a sensitive sequence search method (HHsearch [*Söding, 2005*]). For computational efficiency, an initial scan was done using a single sequence, excluding families where the top hit had an e-value of 1 or greater to any protein of known structure. We identified 100 families satisfying these criteria in *Escherichia coli* (Gram-negative, Proteobacteria), and an additional 22, 5, and 4 families in *Bacillus subtilis* (Gram-positive, Firmicutes bacterium), *Halobacterium salinarum* (Euryarchaeota), and *Sulfolobus solfataricus* (Crenarchaeota), respectively (*Supplementary file 1*). For each of these top families, we carried out a more sensitive profile–profile sequence search against the Protein Data Bank (PDB) using HHsearch (*Söding, 2005*) and the fold recognition method SPARKS-X (*Yang et al., 2011*). We eliminated families if the top HHsearch hit had an E-value less than 1E-04 and was consistent with GREMLIN contacts.

An alternative approach to structure modeling using predicted contacts is to search for weak fold recognition matches to known protein structures and determine if any of the hits fit the predicted contacts. This approach is not very effective for the families identified as described above; for only 4 of the 122 families with HHsearch E-values greater than 1E-04 did one of the top ten hits from HHsearch or SPARKS-X match the predicted contacts (have Rc values greater than 0.6).

Many of the families we identified with no homologs of known structure are transmembrane (TM) proteins. To evaluate the accuracy of our co-evolution-based structure prediction method on TM proteins, we tested it on a benchmark of 13 TM proteins with recently determined structures. Rather than evaluating the lowest energy five models as in the case of the CASP experiment, we instead selected the most central (see 'Materials and methods') low-energy model and eliminated positions not converged within the lowest energy models or not constrained by contact information (see 'Materials and methods'). As shown in *Table 1*, for the 11 of the 13 proteins for which the structure prediction calculations converged, the RMSD of the predicted structure to the experimentally determined structure over the converged and constrained residues was below 4.0 Å (the RMSDs over the structurally aligned regions were all below 2.8 Å). Features such as kinked, discontinuous, and re-entrant helices as well as coils within the bilayer that complicate approaches to membrane protein

**Table 1**. Transmembrane protein benchmark

| PDB | Name | Seq/len | Full protein | | Converged | | Aligned | |
|-----|------|---------|------|--------|------|--------|------|--------|
| | | | rmsd | Length | rmsd | Length | rmsd | Length |
| 4HE8_H (3.3) | NADH-quinone oxidoreductase subunit 8 | 17.3 | 4.9 | 269 | 2.1 | 183 | 2.2 | 234 |
| 1SOR_A (N/A) | Aquaporin-0 | 26.2 | 2.7 | 221 | 2.1 | 188 | 2.0 | 200 |
| 4Q2E_A (3.4) | Phosphatidate cytidylyltransferase | 18.6 | 5.4 | 262 | 3.5 | 176 | 2.8 | 178 |
| 4HTT_A (6.8) | Sec-independent protein translocase protein | 14.6 | 3.9 | 225 | 1.8 | 124 | 2.4 | 181 |
| 4P6V_E (3.5) | Na(+)-translocating NADH-quinone reductase subunit D | 14.3 | 5.0 | 194 | 1.4 | 49 | 2.8 | 155 |
| 4J72_A (3.3) | Phospho-N-acetylmuramoyl-pentapeptide-transferase | 19.9 | 6.6 | 323 | 3.1 | 251 | 2.4 | 237 |
| 3V5U_A (1.9) | Sodium/Calcium exchanger | 10.2 | 3.9 | 297 | 3.7 | 284 | 2.3 | 245 |
| 4PGS_A (2.5) | Uncharacterized protein YetJ | 15.4 | 3.5 | 207 | 2.7 | 175 | 2.2 | 183 |
| 4QTN_A (2.8) | Vitamin B3 transporter PnuC | 9.0 | 4.2 | 202 | 3.0 | 155 | 2.8 | 178 |
| 4OD4_A (3.3) | 4-hydroxybenzoate octaprenyltransferase | 22.8 | 3.9 | 275 | 3.4 | 242 | 2.8 | 231 |
| 4O6M_A (1.9) | CDP-alcohol phosphotransferase | 13.3 | 4.1 | 188 | 4.0 | 165 | 2.3 | 159 |
| 4WD8_A (2.3) | Bestrophin domain protein | 5.94 | N/A | 268 | Not converged | | | |
| 4F35_A (3.2) | Transporter, NadC family | 14.5 | N/A | 434 | Not converged | | | |

Column 1, PDB code (resolution of the crystal structure); column 2, protein name; column 3, sequences per length, after filtering to reduce the redundancy to 90%; column 4, RMSD of predicted structure to native structure; column 5, length of native structure modeled; column 6, RMSD over converged and constrained region; column 7, length of converged and constrained region; column 8, RMSD over TM-align structural alignment; column 9, length of structurally aligned region.

structure prediction that assume the accuracy of a TM helix prediction were all recovered correctly (for example, the re-entrant helices of aquaporin; the power of fragment-based approaches to model such features was noted in *Nugent and Jones, 2012*).

We built models for representatives of the 121 families with unknown structures using the Rosetta co-evolution-guided structure prediction protocol, eliminating from the lowest energy structures the non-converged and non-constrained residues. The calculations converged for 58 of the 121 proteins (*Table 2*). Four targets had Rc values less than 0.7; these targets contain clusters of contacts that may be involved in homo-oligomeric formation. The models are very different from those generated using traditional profile search and threading methods: with the exception of five targets with TMscore of 0.5 (*Table 2*, columns 7–8), the structural similarity of the Rosetta models to the top ranked models generated by HHsearch/SPARK-X is very low. The intractability of modeling these families using profile–profile/fold recognition methods is reiterated by the very low similarity between the models that best fit the contacts produced by HHsearch and SPARKS-X (*Table 2*, column 9; *Supplementary file 2*).

Based on the benchmark, we expect that our monomeric protein models should be within 4.0 Å RMSD of the actual structure. Provided there are not large conformational changes upon docking, protein–protein complexes can be accurately assembled from crystal structures or comparative models of the constituent monomers using GREMLIN contact predictions (*Ovchinnikov et al., 2014*). Thus, the models of complexes we provide in this article are likely to be fairly accurate if the monomeric subunits are predicted accurately, but there is clearly more room for error in our more complex multi-subunit predictions.

The models are available at (External Database: http://gremlin.bakerlab.org/structures/). The biological implications of all of these structures cannot be explored in a single paper; here, we describe functional insights obtained from a subset of the models. These insights derive in part from the distribution of evolutionarily conserved residues in the models, as conserved sequence motifs tend to mark functional sites in structures (*Zuckerkandl and Pauling, 1965*; *Villar and Kauvar, 1994*; *Pei and Grishin, 2001*; *Muth et al., 2012*). As is evident in *Figure 2*, the conserved residues cluster quite strongly in the predicted structures. We describe first, hypotheses on mechanism for proteins of known function, and second, hypotheses on function for proteins with currently unknown function. In the following sections, the predictions are grouped by known biological functions assigned by Clusters of Orthologous Groups (*Galperin et al., 2015*). *Hopf et al. (2012)* also used co-evolution information to guide membrane protein structure prediction and function assignment; we compare to their conclusions in the two cases common to both studies.

## Biological insights from structural models

### Energy production and transport

Cytochrome bd-I ubiquinol oxidase generates a proton-motive force to power the adenosine triphosphate (ATP) synthase when oxygen is limited. The enzyme has two integral membrane subunits (CydA and CydB) with three hemes (heme b595, heme b558, and heme d) that mediate transfer of electrons from quinol to oxygen. Using the co-evolution-guided structure prediction protocol described above, we generated models for the structures of CydA and CydB, and then docked the subunits together using inter-protein predicted contacts as described in *Ovchinnikov et al. (2014)* to generate a model for the entire TM complex (*Figure 3A,B*). The models of CydA and CydB share the same fold—a duplicated four helix bundle unit—and form a pseudo symmetric heterodimer. Structure comparisons of CydA and CydB to the PDB revealed nearly full-length structural similarity to polysulfide reductase TM domain (PsrC) (PDB: 2VPX), an enzyme complex responsible for the quinone-mediated reduction of polysulfide, and structure comparisons for the four helix bundle unit revealed strong similarities to cyt b561 (PDB: 4O7G). The b595 and b558 heme-binding sites of each CydA four helix bundle have been mapped experimentally by mutagenesis: H19 ligates heme b595, and H186 and M393, heme b558. Strikingly, in our model, these residues are aligned with conserved axial ligands in cyt b561 (*Figure 3C*). Residues ligating heme d have not yet been identified experimentally, but in our model, a third conserved CydA histidine, H126 structurally aligns to a known heme-binding site near the cytoplasmic surface of cyt b561. We hypothesize that this residue ligates heme d, which has been proposed to be on the periplasmic side (see *Figure 3C*), a location of heme d near the cytoplasm could explain the proton-motive force generated across the membrane. In addition to the heme-ligating residues, mutagenesis studies (*Borisov et al., 2011*) have identified residues involved in quinone binding and proton flow. In our model of the structure of CydA, the quinone-binding residues (*Figure 3B*, red spheres) cluster

**Table 2.** Comparison of fold recognition and Rosetta models for large protein families

| Known function | | | Rc | | | TMscore | | |
|---|---|---|---|---|---|---|---|---|
| Name | #seq | Ev | HH | SP | M | M_HH | M_SP | HH_SP |
| WECH: O-acetyltransferase (YiaH) | 24,750 | −2.4 | 0.0 | 0.1 | 0.9 | 0.1 | 0.2 | 0.1 |
| SATP: Succinate-acetateproton symporter (YaaH) | 2298 | −2.1 | 0.4 | 0.5 | 1.1 | 0.3 | 0.3 | 0.8 |
| LSPA: Lipoprotein signal peptidase | 8156 | −2.0 | 0.2 | 0.1 | 1.0 | 0.2 | 0.3 | 0.3 |
| YADH: ABC-type multidrug transport permease | 42,626 | −2.0 | 0.1 | 0.1 | 0.7 | 0.3 | 0.2 | 0.2 |
| YEBZ: Putative copper export protein | 4067 | −2.0 | 0.1 | 0.1 | 0.8 | 0.2 | 0.3 | 0.2 |
| CRCB: Fluoride ion exporter | 7829 | −1.8 | 0.2 | 0.3 | 1.0 | 0.2 | 0.2 | 0.3 |
| LPTG: Lipopolysaccharide export system permease | 8101 | −1.8 | 0.0 | 0.1 | 0.9 | 0.1 | 0.1 | 0.2 |
| FTSW: Lipid II flippase | 14,900 | −1.7 | 0.0 | 0.1 | 1.0 | 0.1 | 0.2 | 0.2 |
| RFAL: O-antigen ligase | 13,535 | −1.7 | 0.2 | 0.1 | 0.9 | 0.3 | 0.2 | 0.2 |
| CCMB: Heme exporter protein B | 2433 | −1.6 | 0.1 | 0.1 | 0.7 | 0.2 | 0.2 | 0.2 |
| MLAE: ABC transporter permease for lipid asymmetry | 7662 | −1.4 | 0.0 | 0.1 | 0.9 | 0.1 | 0.2 | 0.3 |
| SULP: Sulfate permease | 6647 | −1.2 | 0.1 | 0.0 | 0.8 | 0.2 | 0.2 | 0.2 |
| TOLQ: Biopolymer transport protein | 9256 | −1.2 | 0.1 | 0.1 | 0.7 | 0.2 | 0.2 | 0.2 |
| LGT: Prolipoprotein diacylglyceryl transferase | 8121 | −1.1 | 0.1 | 0.2 | 1.0 | 0.2 | 0.3 | 0.3 |
| Q97UR7: N-methylhydantoinase B (HyuB-3) | 4491 | −1.0 | 0.1 | 0.1 | 1.1 | 0.1 | 0.1 | 0.1 |
| YGAZ: putative L-valine exporter | 6435 | −1.0 | 0.1 | 0.2 | 0.9 | 0.2 | 0.3 | 0.2 |
| CCMC: Heme exporter protein C | 5965 | −0.8 | 0.1 | 0.1 | 1.1 | 0.2 | 0.2 | 0.2 |
| YEDZ: Sulfoxide reductase heme-binding subunit | 2247 | −0.7 | 0.2 | 0.2 | 1.0 | 0.2 | 0.3 | 0.3 |
| YIAM: TRAP transporter small permease protein | 10,715 | −0.7 | 0.1 | 0.2 | 1.1 | 0.3 | 0.3 | 0.2 |
| TTDA: Tartrate dehydratase, alpha subunit | 4238 | −0.6 | 0.0 | 0.1 | 1.2 | 0.1 | 0.1 | 0.1 |
| UPPP: Undecaprenyl pyrophosphate phosphatase | 7842 | −0.6 | 0.0 | 0.1 | 1.0 | 0.2 | 0.2 | 0.2 |
| PLSY: Probable glycerol-3-phosphate acyltransferase | 6112 | −0.4 | 0.1 | 0.2 | 1.1 | 0.2 | 0.4 | 0.2 |
| FLIL: Flagellar protein | 2690 | −0.3 | 0.7 | 0.5 | 0.8 | 0.5 | 0.4 | 0.9 |
| CYDB: Cytochrome bd oxidase 2 | 6864 | 0.0 | 0.1 | 0.1 | 1.0 | 0.2 | 0.2 | 0.1 |
| CYDA: Cytochrome bd oxidase 1 | 6200 | 0.1 | 0.0 | 0.1 | 1.2 | 0.1 | 0.2 | 0.2 |
| MOTA: Motility protein A, flagellar motor proton conductor | 4734 | 0.3 | 0.1 | 0.1 | 0.9 | 0.1 | 0.1 | 0.2 |
| SLYB: Outer membrane lipoprotein | 1860 | 0.3 | 0.1 | 0.2 | 0.8 | 0.2 | 0.2 | 0.1 |
| MRED: Rod shape-determining protein | 1546 | 0.6 | 0.5 | 0.5 | 0.8 | 0.5 | 0.4 | 0.6 |
| ZUPT: Zinc transporter | 10,517 | 0.6 | 0.1 | 0.1 | 0.8 | 0.2 | 0.1 | 0.2 |
| YOHK: Putative effector of murein hydrolase LrgB | 3941 | 2.3 | 0.2 | 0.1 | 0.9 | 0.4 | 0.2 | 0.2 |
| PRSW: Membrane proteinase | 2500 | 5.3 | 0.2 | 0.2 | 0.9 | 0.3 | 0.3 | 0.7 |
| DDG: Lipid A biosynthesis palmitoleoyl acyltransferase | 9430 | 5.8 | 0.4 | 0.1 | 1.0 | 0.4 | 0.2 | 0.2 |
| Unknown function | | | Rc | | | TMscore | | |
| Name | #seq | Ev | HH | SP | M | M_HH | M_SP | HH_SP |
| YQFA: UPF0073 inner membrane protein | 7596 | −2.6 | 0.1 | 0.4 | 1.1 | 0.2 | 0.5 | 0.3 |
| YCED: Uncharacterized protein | 1604 | −2.5 | 0.1 | 0.2 | 0.9 | 0.2 | 0.2 | 0.2 |
| YPHA: Inner membrane protein | 2986 | −2.2 | 0.1 | 0.4 | 1.0 | 0.2 | 0.3 | 0.2 |
| YADS: UPF0126 inner membrane protein | 5222 | −1.9 | 0.1 | 0.1 | 0.9 | 0.2 | 0.3 | 0.2 |
| YHHN: Uncharacterized membrane protein | 2529 | −1.9 | 0.1 | 0.2 | 0.9 | 0.2 | 0.3 | 0.2 |
| YIDH: Inner membrane protein | 1041 | −1.9 | 0.1 | 0.2 | 0.6 | 0.3 | 0.3 | 0.2 |
| YITE: UPF0750 membrane protein | 8326 | −1.7 | 0.1 | 0.1 | 0.9 | 0.2 | 0.3 | 0.3 |
| HDED: Acid resistance membrane protein | 2885 | −0.6 | 0.1 | 0.2 | 0.8 | 0.2 | 0.2 | 0.2 |

*Table 2. Continued on next page*

*Table 2. Continued*
**Unknown function**

| Name | #seq | Ev | Rc | | | TMscore | | |
|------|------|-----|----|----|----|------|------|------|
| | | | HH | SP | M | M_HH | M_SP | HH_SP |
| YFIP: DTW domain-containing protein | 3100 | −1.5 | 0.2 | 0.2 | 0.9 | 0.2 | 0.2 | 0.1 |
| YPJD: ABC-type uncharacterized permease | 6180 | −1.4 | 0.2 | 0.2 | 0.9 | 0.2 | 0.3 | 0.2 |
| YJFL: UPF0719 inner membrane protein | 1581 | −1.3 | 0.1 | 0.1 | 0.7 | 0.2 | 0.3 | 0.3 |
| YTEJ: Uncharacterized membrane protein | 5733 | −1.2 | 0.1 | 0.1 | 1.0 | 0.2 | 0.2 | 0.2 |
| YIHY: UPF0761 membrane protein | 10,144 | −0.9 | 0.1 | 0.1 | 0.9 | 0.1 | 0.2 | 0.2 |
| YQAA: Inner membrane protein | 2187 | −0.9 | 0.1 | 0.3 | 1.0 | 0.2 | 0.4 | 0.3 |
| YHID: Uncharacterized protein | 4416 | −0.7 | 0.2 | 0.2 | 1.0 | 0.2 | 0.1 | 0.2 |
| YLOU: Uncharacterized protein | 3738 | −0.7 | 0.4 | 0.5 | 0.9 | 0.3 | 0.3 | 0.8 |
| YGDD: UPF0382 inner membrane protein | 3025 | −0.6 | 0.5 | 0.3 | 1.0 | 0.3 | 0.2 | 0.4 |
| YJCH: Inner membrane protein | 1307 | −0.5 | 0.3 | 0.2 | 0.8 | 0.4 | 0.2 | 0.2 |
| YFCA: UPF0721 transmembrane protein | 18,846 | 0.0 | 0.1 | 0.1 | 1.0 | 0.2 | 0.3 | 0.2 |
| YOHJ: Putative effector of murein hydrolase | 3608 | 0.4 | 0.2 | 0.3 | 0.5 | 0.3 | 0.4 | 0.6 |
| YHHQ: Inner membrane protein | 3398 | 0.7 | 0.4 | 0.2 | 1.0 | 0.4 | 0.3 | 0.2 |
| YAII: UPF0178 protein | 3144 | 0.8 | 0.6 | 0.7 | 1.1 | 0.5 | 0.5 | 0.4 |
| YUXK: Predicted thiol-disulfide oxidoreductase | 1881 | 1.3 | 0.3 | 0.3 | 1.1 | 0.3 | 0.3 | 0.5 |
| YICC: UPF0701 protein | 4293 | 1.5 | 0.1 | 0.1 | 1.0 | 0.1 | 0.1 | 0.1 |
| YEIH: UPF0324 inner membrane protein | 4863 | 4.2 | 0.3 | 0.2 | 0.9 | 0.4 | 0.5 | 0.7 |
| RARD: Putative chloramphenical resistance permease | 74,507 | 6.3 | 0.1 | 0.1 | 1.0 | 0.3 | 0.3 | 0.2 |

Column 2: number of unique proteins in family; Column 3: negative log10 of E-value of top match found in HHsearch profile–profile search of PDB; Columns 4–6: fit to predicted contacts (Rc value) of best fitting of top 10 HHsearch hits (column 4), of best fitting of top 10 SPARKS-X hits (column 5), and Rosetta model (column 6). Native structures have Rc values ranging from 0.7 to 1.2 (*Figure 17*). Columns 7–9: structural similarity (TMscore) between Rosetta model (M) and best fitting HHsearch model, between Rosetta model and best fitting SPARKS-X model, and between best fitting HHsearch and SPARKS-X models. The Rosetta models fit the contacts as well as expected for native structures and are very different from best fitting HHsearch and SPARKS-X models. For RARD and YEIH, the HHsearch E-value is less than 1E-04, the recommended threshold for inclusion in the same Pfam clan (*Xu and Dunbrack, 2012*), but the fit with the co-evolutionary contacts was very poor (Rc < 0.3); these two cases are discussed in sections below. For FLIL and YAII, the Rc values for very weak HHSearch and SPARKS-X hits (E-values worse than 0.1) are greater than 0.6 but the contacts constrain only a portion of the structure.

together, and the proton channel residues (*Figure 3B*, blue spheres) cluster together. Two additional conserved residues with no known function (R9 and R448) are near the quinone-binding Q loop on the periplasmic surface. Thus, the CydA model agrees with extensive mutagenesis data and places the cytochrome bd–I complex within the evolutionary context of other TM di-heme cytochromes.

L-tartrate dehydratase is used by *E. coli* under anaerobic conditions to convert L-tartrate (carbon source) to oxaloacetate. The enzyme is a hetero-tetramer, with two copies of TtdA and two copies of TtdB (*Reaney et al., 1993*). TtdA is homologous to the N terminus of a class I fumarase, and TtdB, to the C terminus of the fumarase. The structures of TtdA and the fumarase N-terminus have not been determined, but the structure of the fumarase C-terminal domain has been solved (PDB: 2ISB) and is structurally related to the swiveling domain from aconitase enzymes that perform similar chemistry (*Lauble et al., 1994*). The TtdB-like swiveling domain from aconitase (PDB: 1ACO) binds its substrate near the interface of the swiveling domain and another catalytic domain that binds 4Fe-4S. Given the importance of the adjacent catalytic domain as well as the domain interface in aconitase, we predicted the structure of TtdA and assembled it into the hetero-tetramer complex with a homology model of TtdB (*Figure 4A*). In our complex model, three conserved TtdA cysteines (C71, C190, and C277) cluster near the TtdB interface which maybe the 4Fe-4S cluster-binding site (*Figure 4B*). This potential active site also includes a conserved aspartate (D73) that might contribute to catalysis. The conserved TtdB H265 falls on the opposite side of the active site in our model and instead contributes to the active site of the second TtdB chain formed by the tetramer. Thus, our model suggests TtdA/TtdB forms an obligate tetramer that would not have been predicted by co-evolution or conservation alone.

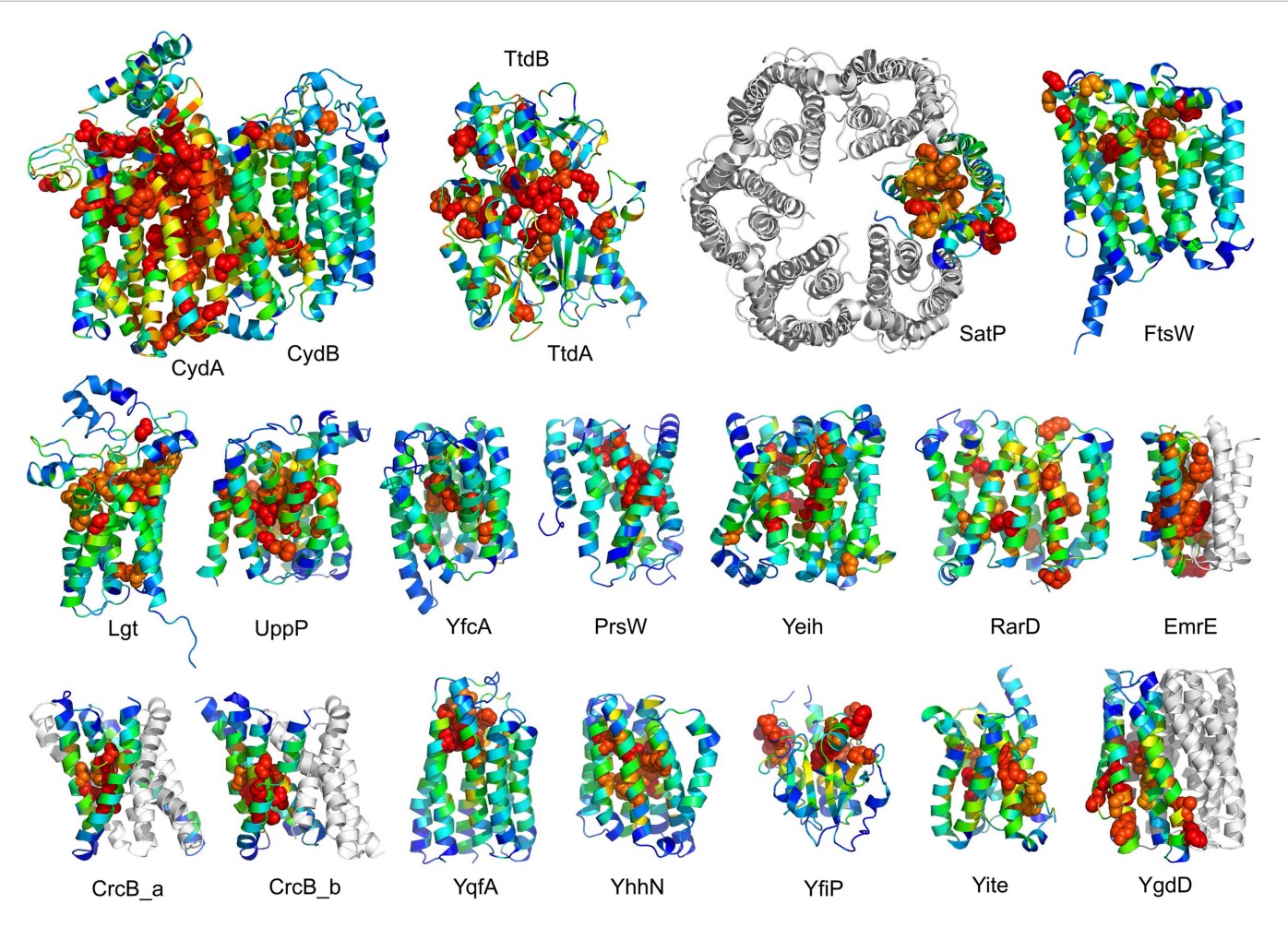

**Figure 2**. Conserved residues tend to cluster in the predicted structures. Residue conservations from multiple sequence alignments were mapped to predicted structures using Al2Co (*Pei and Grishin, 2001*) and are colored in rainbow from blue (variable) to red (conserved). The most conserved residues (red or orange), displayed as spheres to highlight their positions, tend to line interaction surfaces and indicate potential functional sites.

SatP (Succinate-acetate/proton symporter) mediates the uptake of succinate and acetate in *E. coli* coupled to proton symport (*Sá-Pessoa et al., 2013*). Our predicted SatP structure (*Figure 5A–C*) is very similar to that of the proton-gated urea channel (*Figure 5D*). The urea channel assembles into a hexameric ring with each protomer forming a channel through the center of the 6TMH fold. Conserved residues line both the channel and the protomer interface and are important for proton gating and solute selectivity. Assembly of our SatP model into a hexameric ring satisfied predicted contacts not made in the monomer (*Figure 5A,B*). Residues that have been shown to influence the solute selectivity of SatP (Leu131 and Ala164) (*Sá-Pessoa et al., 2013*) line the channel pore of our model (*Figure 5C*). Most of the conserved SatP residues line the channel at a similar depth as the constriction sites in UreI and are likely involved in similar gating and selectivity functions as their UreI counterparts. A cluster of conserved residues face the periplasmic surface and align to the periplasmic loop (PL1) in UreI that is thought to plug the channel in a proton-dependent manner (*Hommais et al., 2004*). The similarity of our model to the SatP fold is not only supported by mutagenesis data but also suggests the functional importance of multimeric assembly not revealed by co-evolution or conservation analysis alone.

## Lipid and bacterial cell wall synthesis

Bacterial cell wall synthesis involves multiple steps. FtsW is an integral membrane protein that is thought to transfer lipid-linked peptidoglycan precursors from the inner to the outer leaflet of the

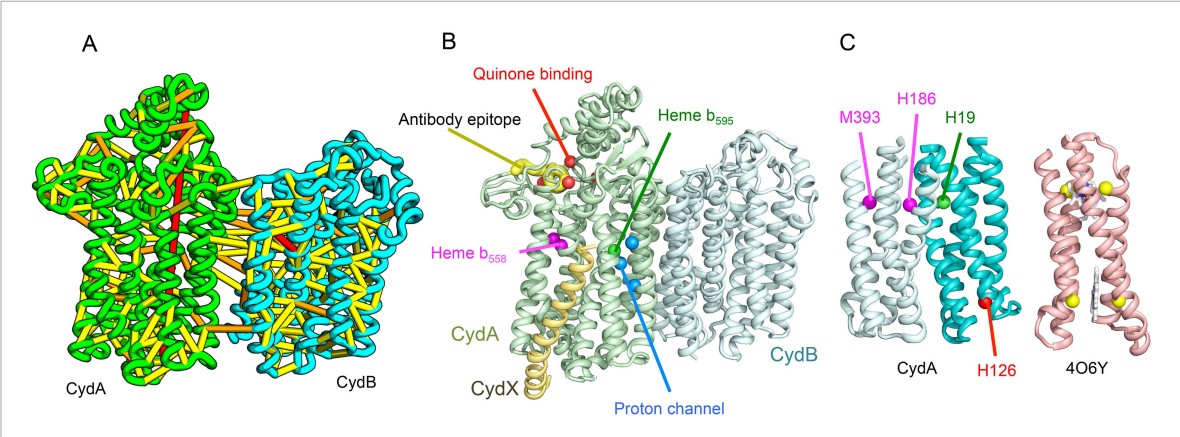

**Figure 3**. Predicted structure of the Cytochrome bd oxidase complex. (**A**) Location of the top co-evolving residue pairs in our model. For clarity, the monomers have been pulled apart slightly. (**B**) Location of conserved and experimentally characterized residues (*Borisov et al., 2011*) on structure model. (**C**) Residues that coordinate heme in CydA are in the same location as histidines (yellow spheres) in Cytochrome b561 (PDB: 4O6Y). H126 (red sphere) overlaps one of these histidines and is the proposed as a heme d coordination site. For clarity, both the model of CydA and the structure of Cyt b562 (4O6Y) are trimmed to highlight the four helix bundle(s).

cytoplasmic membrane, where it interacts with the TM portion of peptidoglycan synthetase FtsI (*Fraipont et al., 2011*). Using co-evolutionary information for the FtsW family, and between it and FtsI, we generated a model of FtsW in complex with the TM domain of FtsI (*Figure 6A,B*). The FtsW model encompasses 10 TM helices, with the last seven (TMH4-TMH10) adopting a similar topology as TMH4-TMH7 and TMH10-12 of the TM domain of STT3 (PDB: 3WAK) (*Figure 6C,D*). STT3 is a dolichyl-diphosphooligosaccharide-protein glycosyltransferase that functions in N-glycan biosynthesis, transferring oligosaccharides from the membrane anchor dolichol-diphosphate to asparagine residues of proteins bound for secretion (*Matsumoto et al., 2013*). The FtsW substrate, lipid II, has a membrane anchor similar to that of the STT3 substrate donor: bactoprenol-pyrophosphate conjugated to disaccharide. A conserved DxH motif at the N-terminus of STT3 TMH4 coordinates a divalent metal ion in the active site. Two residues from the corresponding TMH4 of FtsW (R145 and K153) are essential for flippase activity, with the side chain of R145 overlapping the divalent metal in superpositions of the FtsW model with the STT3 structure. Other conserved FtsW residues line this site and probably contribute to function. The conserved FtsW/STT3 TMH core is similar in sequence to the *E. coli* O-antigen ligase RfaL, and our predicted structure for RfaL is similar in structure (*Figure 6E*). Thus, the structure model of FtsW suggest a potential active site analogous to that of the structurally related STT3 TMH core and unites the family with another bacterial cell wall biogenic enzyme.

*E. coli* prolipoprotein diacylglyceryl transferase (Lgt) is an inner membrane protein that transfers the diacylglyceryl moiety from phosphatidylglycerol to an N-terminal cysteine residue that follows the signal peptide of prolipoproteins. Our predicted structure of Lgt has a novel seven trans-membrane helix (TMH) fold, with many of the conserved residues (Y26, R134, N146, E151, G154, R239, and E243) clustering near the proposed periplasmic surface to form a putative active site (*Figure 7*, white spheres); the activity of Lgt is lost or greatly reduced upon mutating these residues to alanine (*Pailler et al., 2012*). The topology and orientation of the TMHs in our structural model are consistent with a previously proposed topology model (*Pailler et al., 2012*).

UppP (undecaprenyl pyrophosphate phosphatase), an integral membrane protein with unknown structure, catalyzes the dephosphorylation of undecaprenyl pyrophosphate to form undecaprenyl phosphate, an essential carrier lipid for bacterial peptidoglycan cell wall synthesis (*El Ghachi et al., 2005*). In our UppP structure model, TMH1 and TMH5 form broken helices that enter and exit the membrane on the same side, placing both catalytic regions near to the core of the structure (*Figure 8A*). In contrast to a previously proposed model that assumed unbroken helices (*Chang et al., 2014*), our model has a twofold symmetry between the broken TMH1-TMH4 and the broken TMH5-TMH8 that is mirrored by an internal duplication present in the UppP sequence. A similarly

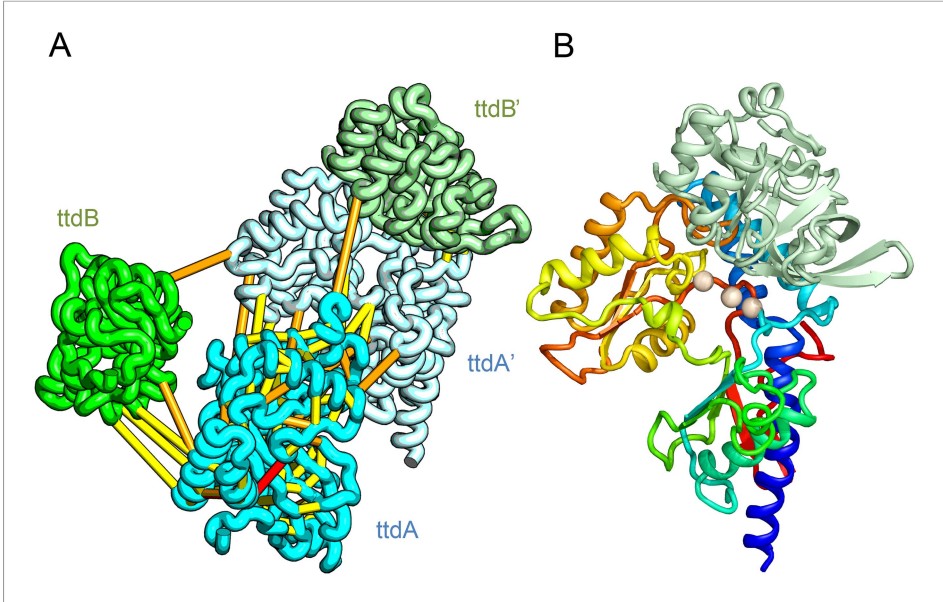

**Figure 4**. Predicted structure of the tartrate dehydratase heterotetramer composed of two copies each of ttdA and ttdB. (**A**) Co-evolving residue pairs. The monomers have been pulled apart to reveal the contacts. (**B**) The ttdA subunit (rainbow) contains a 4Fe-4S cluster (white spheres) that is near the interface with ttdB (green).

duplicated TMH family of unknown function (YfcA) is distantly related to UppP by sequence. Our structure prediction calculations suggest that YfcA has the same fold (*Figure 8B*; the multiple sequence alignments used to make the predictions for UppP and YfcA do not share any sequences). The UppP model illustrates the ability of our method to model unusual structural features such as the broken TMH helices, which are typically difficult to model without precedence in existing structure templates.

## Proteases

PrsW of *B. subtilis* is an intramembrane protease that cleaves site-1 anti-σ factor RsiW, a crucial step in the resistance to antimicrobial peptides (*Ellermeier and Losick, 2006*). PrsW belongs to a large

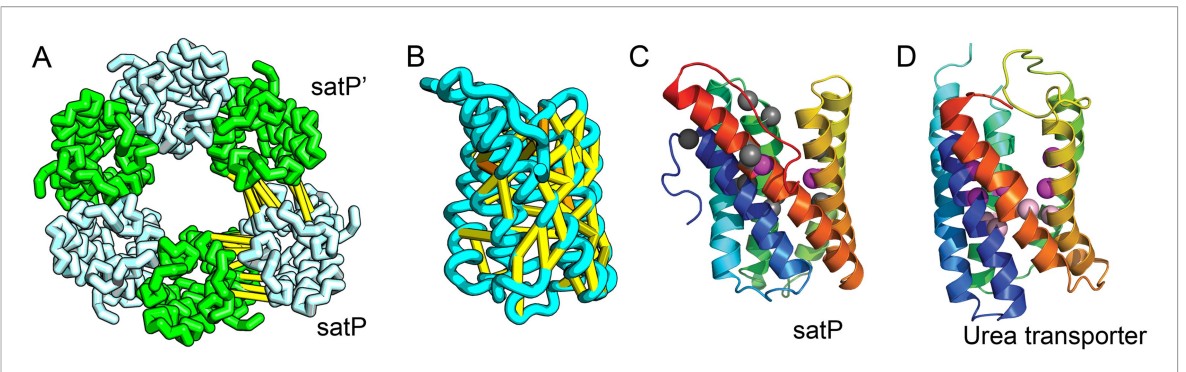

**Figure 5**. Succinate-acetate/proton symporter SatP (YaaH). (**A**) Co-evolving residue pairs in homo-oligomer model. (**B**) Co-evolving residue pairs in SatP monomer model. (**C**) SatP co-evolution-based model places known acetate selective residues (magenta spheres) lining the channel. Conserved residues (gray spheres) line the periplasmic surface. The 6TMH channels are formed by threefold pseudo-symmetric TMH hairpins. (**D**) Proton-gated UreI channel protomer. C-alpha positions at the periplasmic constriction site (magenta spheres) and the cytoplasmic constriction site (pink spheres) are highlighted. The SatP model has the same fold as UreI (**C** vs **D**).

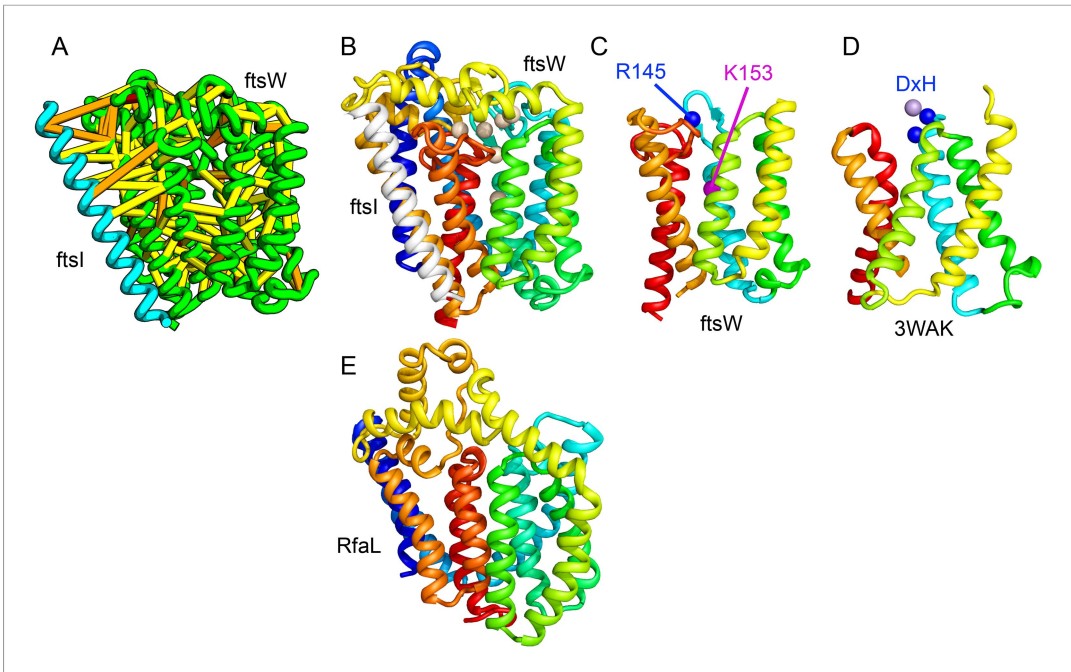

**Figure 6**. Lipid II flippase (FtsW) in complex with the transmembrane domain of Peptidoglycan synthase (FtsI). FtsW is an essential cell division protein that transports lipids across the cytoplasmic membrane and is required for localization of FtsI. (**A**) Location of the top co-evolving residue pairs. (**B**) White spheres indicate conserved positions in FtsW that when mutated to alanine result in loss of flippase activity. (**C**, **D**) The last seven transmembrane (TM) helices of FtsW (TMH4-TMH10) adopting a similar topology as TMH4-TMH7 and TMH10-12 of the TM domain of STT3 (PDB: 3WAK). Both the model of FtsW and 3WAK was trimmed over the aligned helices for clarity. (**C**) Two residues from the corresponding TMH4 of FtsW (R145 and K153) are essential for flippase activity. (**D**) The side chain of R145 overlaps the residues that coordinate the divalent metal in the conserved DxH motif at the N-terminus of STT3 TMH4. (**E**) The model of RfaL adopts a similar fold as ftsW.

superfamily of membrane proteins that includes putative bacteriocin-processing enzymes and the APH-1 subunit of gamma-secretase (*Pei et al., 2011b*). Our PrsW model has structural similarity to an archaeal type II CAAX prenyl protease (*Manolaridis et al., 2013*), mostly in a core of four TMHs (TMHs 3–6 in PrsW model and TMHs 4–7 in type II CAAX prenyl protease) (*Figure 9C,D*). The predicted active site residues in motifs EExxK (TMH3) and HxxxD (TMH6) of PrsW occupy structurally compatible positions as conserved residues in motifs EExxxR (TMH4) and HxxxN (TMH7) of type II CAAX prenyl protease. Another conserved histidine in the fifth TMH of PrsW (but absent in type II CAAX prenyl protease) is also located in the predicted active site of PrsW.

## Transporters

The inner membrane protein YeiH is classified as a member of the CPA/AT transporter clan in PFAM and sequence search yields high confidence matches to sodium/proton antiporters (HHsearch e-value 2.2E-05). Remarkably, although our structural model of YeiH (*Figure 10A,B*) superimposes structurally with the structure of the antiporter NapA (PDB: 4BWZ; *Lee et al., 2013*), the connectivity of the core of the structure is completely different. The core domain of NapA contains two antiparallel discontinuous helices (TM4a, 4b and TM11a, 11b) that cross over each other (*Figure 10C*). In our YeiH model, the same hourglass-shaped assembly is formed by two pairs of broken helices (TM5, 6 and TM8, 9) that exit the membrane on the same side (*Figure 10B*). Discontinuous helices have been found in several transporter proteins and are frequently involved in ion binding (*Screpanti and Hunte, 2007*); a similar arrangement of broken helices is also observed in structures of CLC chloride channels and glutamate transporter Glt (*Dutzler et al., 2002*; *Yernool et al., 2004*). Like the UppP example, our model of YeiH highlights differences in the TM core that are hard to model with template-based homology modeling approaches.

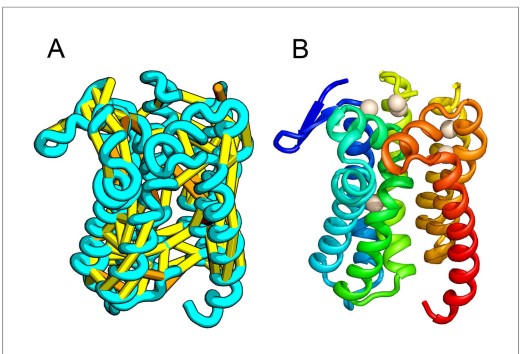

**Figure 7**. Prolipoprotein diacylglyceryl transferase (LGT). (**A**) Predicted contacts indicated on model, (**B**) model with conserved positions at which alanine mutations result in loss in activity indicated in white spheres; five of these are clustered at the periplasmic end of the model.

The *E. coli* chloramphenical resistance protein RarD is an apparent duplication of the homodimeric *E. coli* EmrE drug transporter, and our predicted structure of RarD indeed adopts an internal pseudo-symmetric fold. Our RarD model is structurally superimposable on the EmrE homodimer, but the first helices of the domains corresponding to the EmrE monomer are swapped (*Figure 11B*) The duplicated domains in RarD (called EamA domains) differ from EmrE by a critical helix insertion between helix-1 and helix-2 that causes helix-1 to adopt an inverted conformation in the membrane. The only way helix-1 in EamA can preserve the interactions seen in EmrE is to instead interact with the second copy of EamA (which is inverted in the membrane) as in our structure model (*Figure 11C*). The EamA protein, also composed of two EamA domains, was previously modeled (*Hopf et al., 2012*) but no structural similarity was reported to EmrE.

EmrE is one of a small number of dual-topology TM proteins in which a single polypeptide chain can insert into the membrane in two opposite orientations, thus yielding inverted symmetric TMH topologies (*Duran and Meiler, 2013*). This inverted symmetry is fixed in the monomeric RarD structure. The proposed transport mechanism for EmrE involves switching the dimeric structure between alternate access states (*Fleishman et al., 2006*; *Morrison et al., 2012*). The homologous relationship between EmrE and RarD suggests the inverted symmetric RarD structure might also adopt alternate access states involving the two duplicated EamA halves. Our set of predicted structures includes a member of a second predicted dual-topology protein, the dimeric fluoride transporter CrcB (*Rapp et al., 2006*), which adopts the dual topology (see External Database).

Internal pseudo-symmetry is often observed in the structures of TM proteins. Evolutionary pathways leading to such symmetry can involve gene duplication and fusion events, this is particularly likely when the symmetric single-chain protein has the same overall fold as a known homo-oligomer. While these duplication events can be revealed by the presence of internal sequence repeats, the tendency of the duplicated sequence to diverge and adopt alternate or specialized functions can mask detection of duplication events at the level of primary sequence. The sparseness of determined TM protein structures further complicates analysis of evolutionary folding pathways (*Duran and Meiler, 2013*). Our co-evolution-based structure models substantially increase known TM protein fold space (many have TMalign [*Zhang and Skolnick, 2005*] scores <0.5 to any known structure, *Supplementary file 2*), populating it with new structures that reveal evolutionary folding pathways.

## Unknown function

Protein structures with similar topology often have similar function. Building models allows detection of fold similarity to previously solved structures in the absence of significant sequence homology.

Our models of *E. coli* proteins YqfA and YhhN have topologies similar to that of G-protein-coupled receptors (GPCRS). The YqfA sequence belongs to a large family of integral membrane proteins, with members in all three kingdoms of life. The eukaryotic members are seven-TM pass receptors for ligands such as prostaglandin and adipoQ (the progesterone-adiponectin receptor (PAQR) was predicted to be bacteriorhodopsin-like by *Hopf et al., 2012*), while a bacterial member is associated with furfural tolerance through an unknown mechanism. The PAQR receptors belong to a larger superfamily of core seven TM-bound putative hydrolases identified as CREST (*Pei et al., 2011a*). The CREST superfamily is characterized by conserved motifs at the end of TMH2 (SxxxH), the beginning of TMH3 (D), and the beginning of TMH7 (HxxxH). In the YqfA model, the conserved CREST motifs likely form an active site and are in the same region as the ligand-binding pocket in GPCRs (*Figure 12*). During the preparation of this manuscript, the structure of a homolog of YqfA was released by the PDB; our model of YqfA is very similar to this structure (TMalign score of 0.80; *Figure 12E*), while the

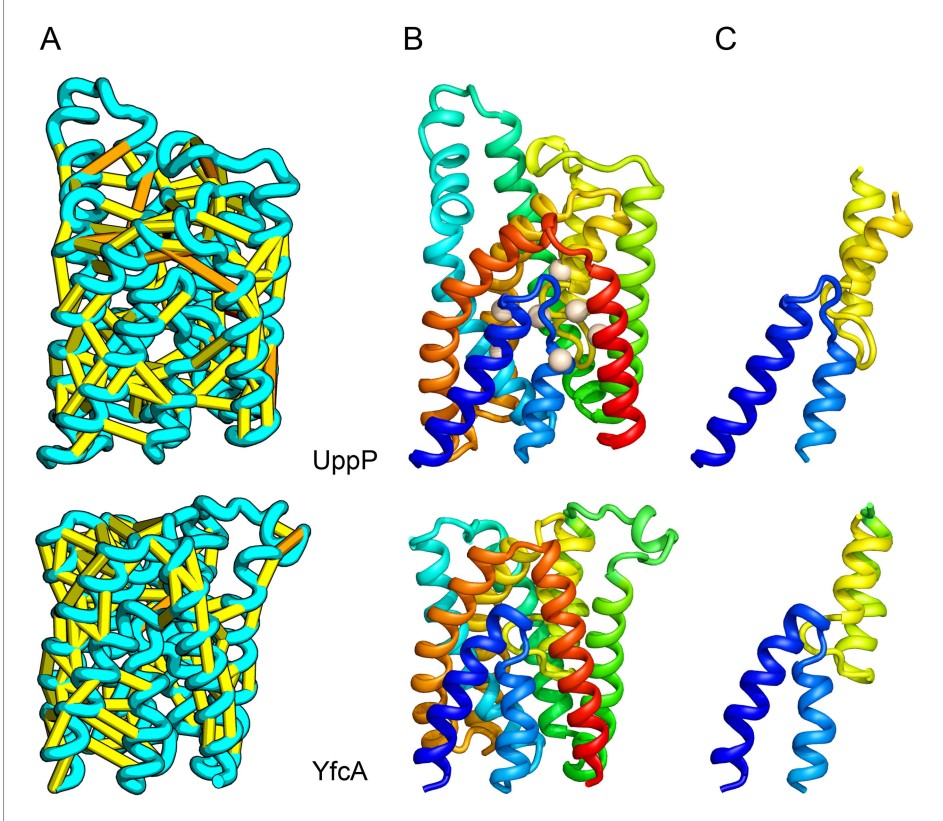

**Figure 8**. UppP catalyzes the dephosphorylation of undecaprenyl diphophate (UPP). (**A**) Location of the top co-evolving residue pairs. (**B**) Spheres in white indicate conserved residues experimentally shown to decrease activity to <1% (*Chang et al., 2014*); all these residues are in the core in the model. YfcA, a protein of unknown function is a very distant sequence homologue of UppP (they are in different PFAM families); (**C**) the predicted structure of YfcA has the same fold as UppP with prominent broken helices (highlighted in blue and yellow).

top hit of HHsearch and SPARKS-X models are not similar (TMalign score of 0.21 and 0.40). Our co-evolution-based model of YhhN (*Figure 12C*) has the GPCR topology, but with an N-terminal TMH extension. A YhhN family member was recently shown to function as a lysoplasmalogenase that catalyzes hydrolysis of the vinyl ether bond of lysoplasmalogen in lipid metabolism. Conserved YhhN residues that might form the active site cluster in a similar place as the YqfA putative catalytic residues, lining what would be the ligand-binding site in GPCRs (*Figure 12C,D*).

Our model of YfiP, from a family of unknown function, contains a non-trivial trefoil knot topology characteristic of the alpha/beta knot methyltransferase (SPOUT) superfamily (*Anantharaman et al., 2002*). SPOUT structures utilize conserved residues to bind the AdoMet substrate in a binding cleft formed by the knot. In the predicted YfiP structure, conserved DTW domain residues surround the AdoMet binding cleft, including Asp113, Thr115, Trp116, Pro87, Tyr145, Arg148, Thr158, and Glu160 (*Figure 13*). The predicted similarity of YfiP to the SPOUT methyltransferase fold substantiates a previously suggested role in rRNA processing (*Burroughs and Aravind, 2014*), as rRNA maturation requires extensive nucleotide modifications. The YfiP model is an example of a prediction of an unusual structure that is indicative of function and cannot be predicted by co-evolution or conservation alone.

Our model of YitE, from a protein family of unknown function, has an arrangement of secondary structures nearly identical to the aquaporin water channel fold (*Figure 14*), including the pseudo-symmetric repeat unit, but with completely different connectivity. The two half-helices that meet at the center of the protein are a key feature of water channels and critical for proton exclusion (*Gonen et al., 2005*). The YitE model does not have the 'NPA' motif in the half helix characteristic of water channels, but one of the half helices has an N pointing into the putative pore.

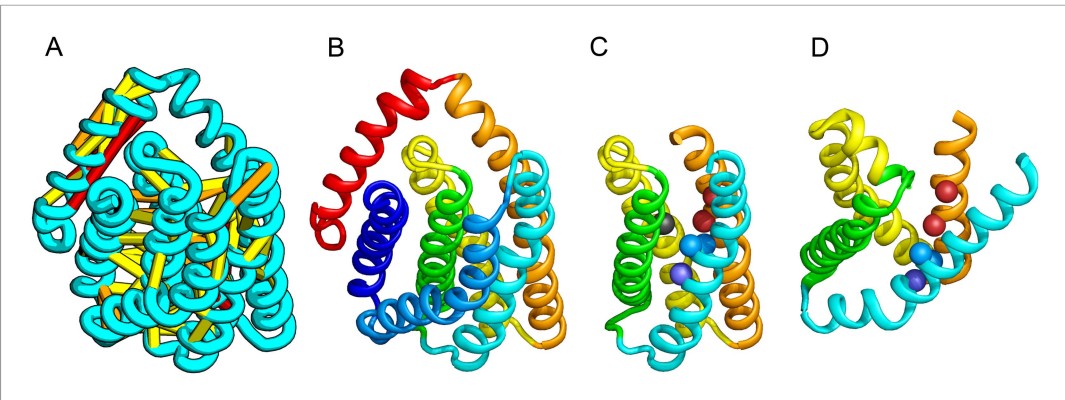

**Figure 9**. PrsW is an intramembrane protease that is crucial in the resistance to antimicrobial peptides. (**A**) Location of the top co-evolving residue pairs. Our model of PrsW (**B**) contains a 4TMH substructure ([TMHs 3–6], (**C**) which is very similar to a substructure of type II CAAX prenyl protease [TMHs 4–7; **D**]). The predicted active site residues in PrsW motifs EExxK (TMH3; blue spheres, **C**) and HxxxD (TMH6; red spheres, **C**) are in positions similar to those of conserved residues in motifs EExxxR (TMH4; blue spheres, **D**) and HxxxN (TMH7; red spheres, **D**) of type II CAAX prenyl protease. Another conserved histidine in the fifth TMH of PrsW (but absent in type II CAAX prenyl protease) is also located in the predicted active site of PrsW (black sphere).

The *E. coli* YgdD protein belongs to a family of unknown function (Pfam: DUF423) with members widely distributed in both bacteria and eukaryotes, including proteins from plants, fungi, and metazoans. The predicted contacts of YgdD are best accommodated in a homotrimer model (*Figure 15A,B*). Each YgdD molecule in the homotrimer has four TMHs with left-handed connections between helices 1, 2, 3 as well as between helices 2, 3, 4. Such a topology matches that of Membrane-Associated Proteins in Eicosanoid

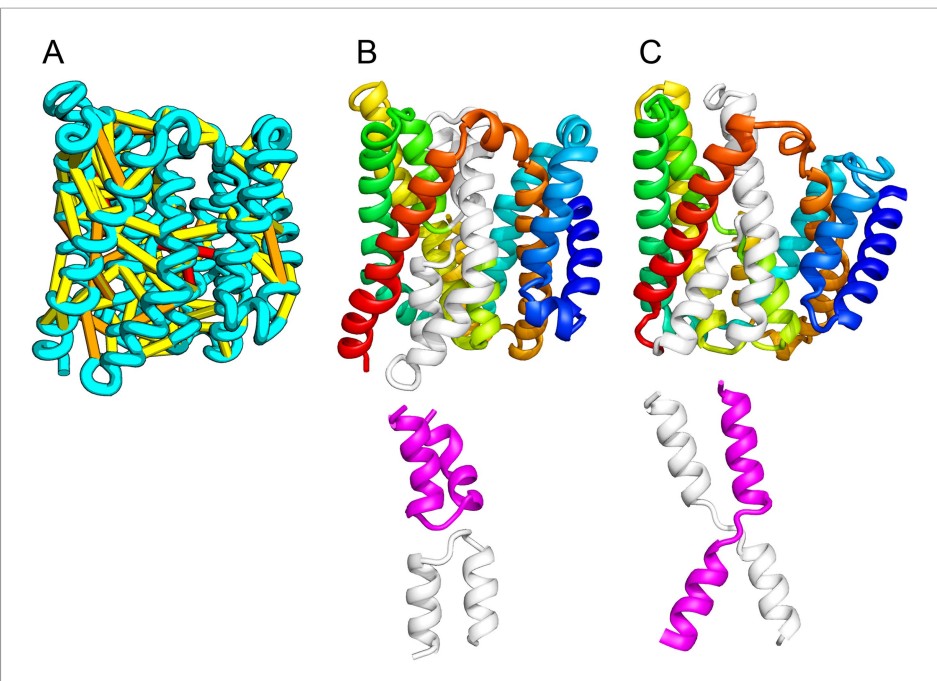

**Figure 10**. Our model of the inner membrane protein YeiH (**A**, **B**) is structurally similar to the structure of the antiporter NapA (**C**). Lower panels: TM helices of core domains are highlighted in white and magenta: while these helices cross over each other in NapA (right), the core of our model of YeiH (left) is formed by two pairs of broken helices (TM5, 6 and TM8, 9) that exit the membrane on the same side.

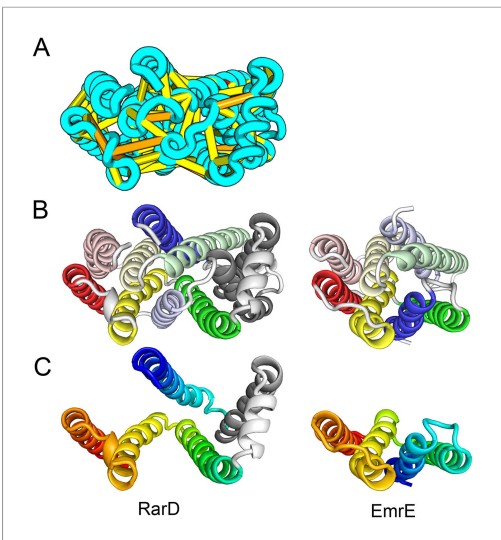

**Figure 11**. Our model of RarD has a similar architecture to EmrE but different fold. (**A**, **B**) Full-length RarD and EmrE homodimer. (**C**) RarD internal repeat and EmrE monomer. The N-terminus helix (blue) is swapped in RarD relative to EmrE due to helix insertion (gray).

and Glutathione metabolism (MAPEG) family proteins (*Jakobsson et al., 1999*; *Hebert and Jegerschöld, 2007*). More strikingly, YgdD and MAPEG also exhibit the same overall homotrimer topology, both similar to the core of the heme-copper oxidase catalytic subunit (*Pei et al., 2014*) (*Figure 15C,D*). Sequence similarity searches of YgdD by HHsearch (*Remmert et al., 2012*) did not reveal significant hits to MAPEG proteins, but weak HHsearch matches to heme-copper oxidase members were found (e.g., 3mk7, chain A, HHsearch probability score: 41). The sequence alignment between the last two TMHs of YgdD and the last two TMHs of heme-copper oxidase members is consistent with the structural alignment between our predicted YgdD structure and the 3mk7 structure, suggesting that YgdD is evolutionarily related to heme-copper oxidases.

## Discussion

The models presented in this article for 58 large protein families which cannot be accurately modeled using comparative modeling or fold recognition methods cover a significant fraction of the prokaryotic sequences for which structural information was previously unavailable. Each of these families have thousands of members (*Table 2*, column 2), hence these models have quite broad impact. The analyses of a small subset of these structures provided here only begins to uncover the wealth of information relating to function they contain. In addition to the new structure-based interpretation of existing sequence conservation and mutational data the models enable, they illustrate the complex transformations occurring in membrane protein evolution: for example, the changes in YeiH and RarD structural element connectivity compared to previously known structures. With the advent of sensitive sequence profile–profile comparison methods, much of protein structure modeling has been reduced to sequence alignment, and indeed for functional interpretation often much can be learned simply by draping the query sequence on a homologous structure; in contrast, as illustrated in the examples above, in the co-evolution-guided de novo structure prediction case, structure modeling is critical to functional insight.

Large-scale genome sequencing is having an unanticipated impact on protein structure modeling, enabling accurate protein structure and protein complex modeling using co-evolution-based predicted contacts. The importance of this approach to structural biology over the next decade will depend on the balance between two opposing trends: as more organisms are sequenced, the number of protein families with sufficient sequences for accurate modeling will increase, but as more structures are determined, there are fewer families for which accurate models cannot be produced by reliable comparative modeling methods. An increase in the number of eukaryotes sequenced—for example, by projects such as the recent Tara Ocean expedition (*Bork et al., 2015*; *Sunagawa et al., 2015*)—would make it possible to accurately model a large number of eukaryote-specific protein families of considerable biological interest. Because of the comparative difficulty in experimental structure determination, it is likely that co-evolution-based prediction will continue to have the most impact for membrane proteins.

In this article, we present models for half of the large protein families in prokaryotes which do not currently have structures. The value of a comparable number of structures of eukaryotic protein families may justify the investment in genome sequencing of a diverse set of ~400 simple eukaryotes. For proteins not belonging to sufficiently large or diverse families but for which functional selections have been developed, it should be possible to develop experimental sequence covariation data sets by library generation, functional selection, and next generation sequencing. Significant resources were invested in the Protein Structure Initiative (PSI), with the initial goal 'to make the three-dimensional, atomic-level structures of most proteins easily obtainable from knowledge of their

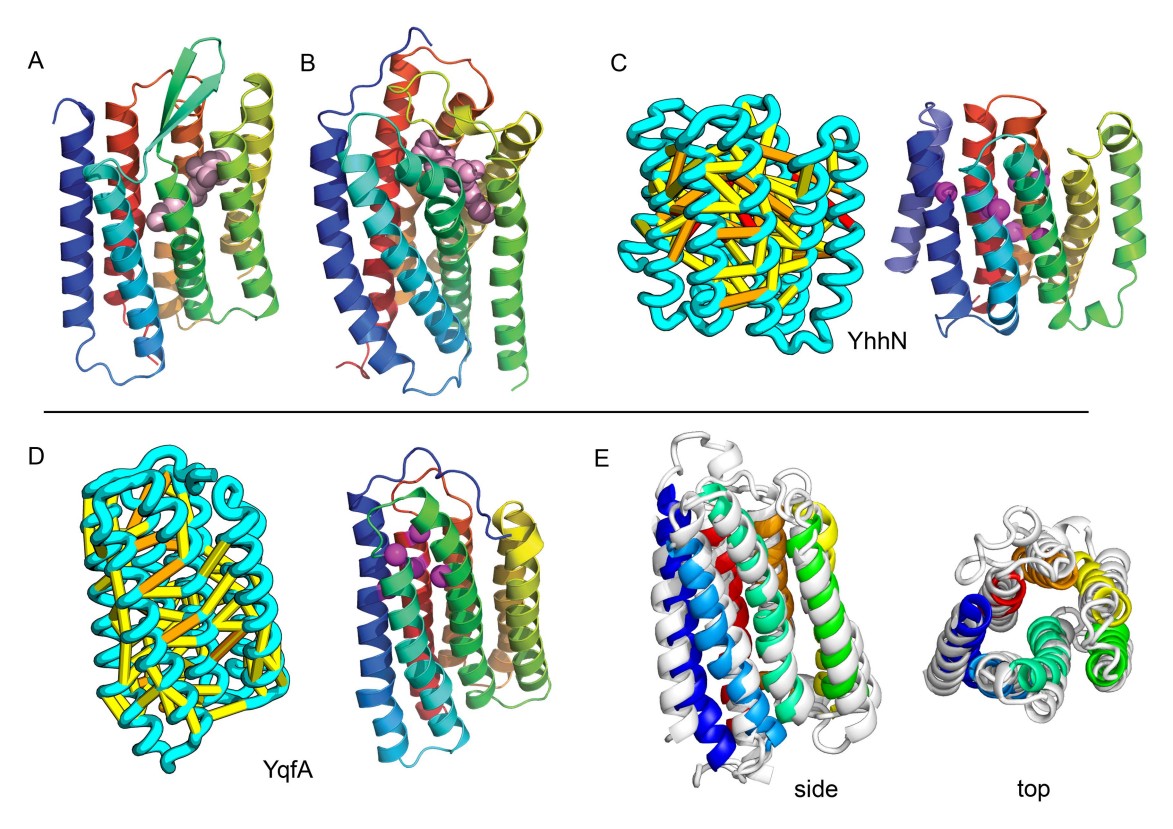

**Figure 12**. Predicted structures of YqfA and YhhN have topologies similar to G protein-coupled receptors (GPCR-like). The core seven TM helix (TMH) fold exhibited by members of the GPCR superfamily is colored in rainbow from the N- to the C-terminus. (**A**) Bacteriorhodopsin binds retinal (pink spheres) in a pocket formed by TMH3-7 [PDB ID: 1m0k]. (**B**) The agonist (pink spheres) binding site of P2Y12 receptor is formed by the same set of helices [PDB ID: 4pxz]. (**C**) A co-evolution-based structure model for YhhN has the GPCR topology with an N-terminal TMH extension. Conserved residues that might form an active site (magenta spheres) cluster in a similar place as the YqfA catalytic residues. (**D**) Our co-evolution-based structure model for YqfA has a GPCR like topology and clusters residues that may form an active site (magenta spheres mark the Calpha position) in a region that corresponds to the GPCR ligand-binding pocket. (**E**) Side and top view of the TMalign superposition of YqfA model (in rainbow) over the recently released 3wxw (in white) human ortholog. The N- and C-terminal loops were trimmed for clarity. The TMalign score between the model and the homolog is 0.8.

corresponding DNA sequences (*Burley et al., 2008*)'. It is notable that structure models can now be generated for exactly the original class of proteins targeted by the PSI—large protein families without any available information—but at a small fraction of the cost.

## Materials and methods

### Multiple sequence alignment generation

Protein-coding genes were extracted from the *E. coli* (AUP000000625), *B. subtilis* (AUP000001570), *H. salinarum* (AUP000000554), *S. solfataricus* (AUP000001974) reference genomes in the UniProt proteome database (*UniProt Consortium, 2014*). Each protein from these proteomes was scanned against the PDB using HHsearch (-ssm 0, from HHsuite v. 2.0.15; [*Remmert et al., 2012*]) to identify proteins with no homologs of known structures (e-value of the top hit >1). We used two versions of the PDB database, one from 01 January, 2012 and one from 31 January, 2015. For the subset that had no hits in 2015, a multiple sequence alignment (MSA) was generated using Jackhmmer (-E 1E-20 -N 8, [*Eddy, 2009*]) and the uniref90 database (*Suzek et al., 2007*) from January, 2015. The alignments were filtered using HHfilter (-id 90 -cov 75), and positions that had more than 75% gaps were removed. To reduce redundancy, we constructed hidden Markov models (HMMs) using HHmake from each MSA and clustered the HMMs based on HHΔ (*Kamisetty et al., 2013*), a measure of HMM–HMM similarity. Families were assigned to

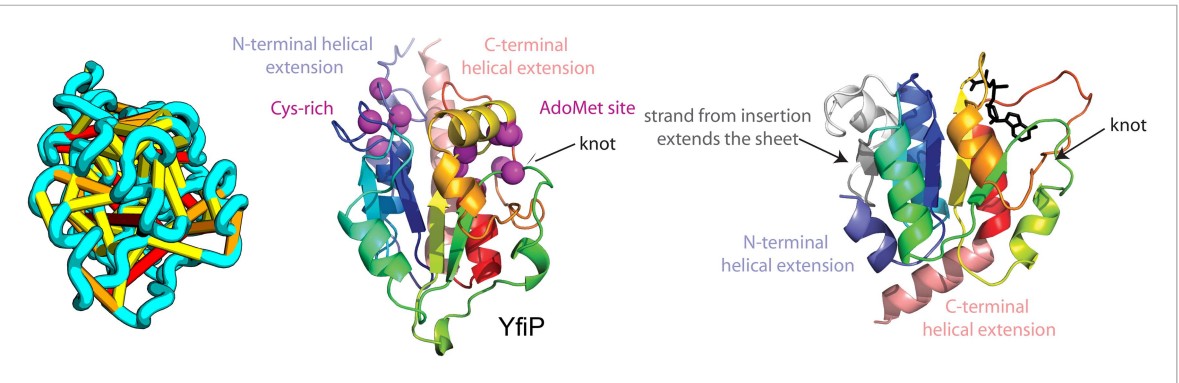

**Figure 13**. YfiP predicted structure has methyltransferase-like fold with knot. Left: the top co-evolving residues pairs. Middle: conserved residues (magenta) surround the AdoMet-binding site and a conserved Cys could bind a Fe4S cluster. Right: 3nk7 methyltransferase bound to AdoMet.

the same cluster if the HHΔ was less than 0.5. The shortest *E. coli* protein was selected in each cluster; if no *E. coli* protein was in the cluster, a representative from *B. subtilis*, *H. salinrum*, or *S. solfataricus* was selected. Families for which the (number of sequences)/(length of representative protein) were greater than four were selected for modeling as described below. If the GREMLIN-predicted contacts (see below) were sparse and primarily between residues close along the linear sequence, the alignment was regenerated at an e-value 1E-40 cutoff, and the GREMLIN calculation repeated. If this resulted in too few sequences, the family was discarded (this eliminated six families).

TM protein domains that had a hit (e-value < 1E-20) in 2015, but no hit (e-value > 1) in 2012 were selected for the TM benchmark. Alignments were created using the UniProt sequence associated with the PDB, and trimmed at the N and C termini to match the crystal structure. We also include aquaporin (PDB: 1SOR_A) to test our protocol in modeling reentrant helices.

## Contact prediction
GREMLIN (v2.01) was used to learn a global statistical model of the sequences in large families using pseudolikelihood optimization (*Balakrishnan et al., 2011*). We previously reported that the accuracy

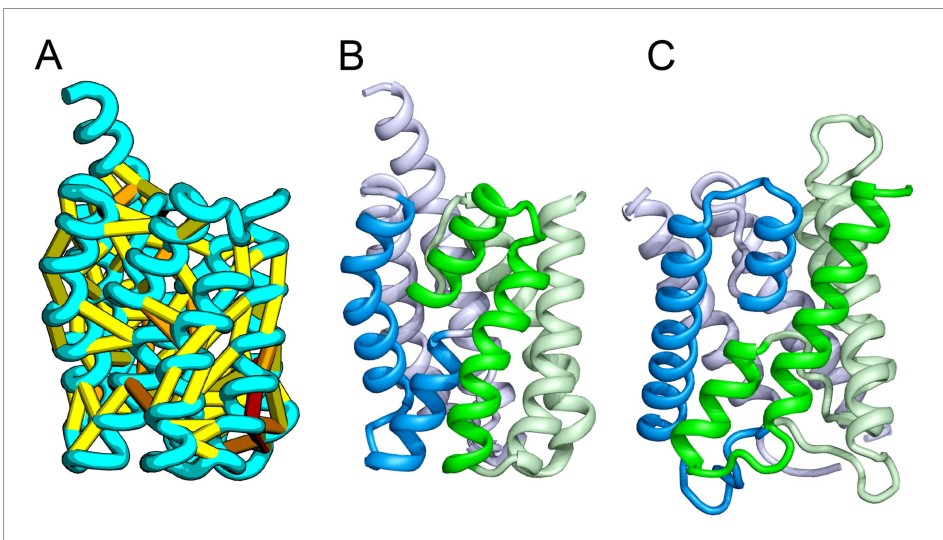

**Figure 14**. *Bacillus subtilis* YitE model. (**A**) The top co-evolving residues. YitE (**B**) has architecture similar to aquaporin (PDB:2B6P) (**C**), including the internal pseudo-symmetry (blue vs green), but completely different connectivity.

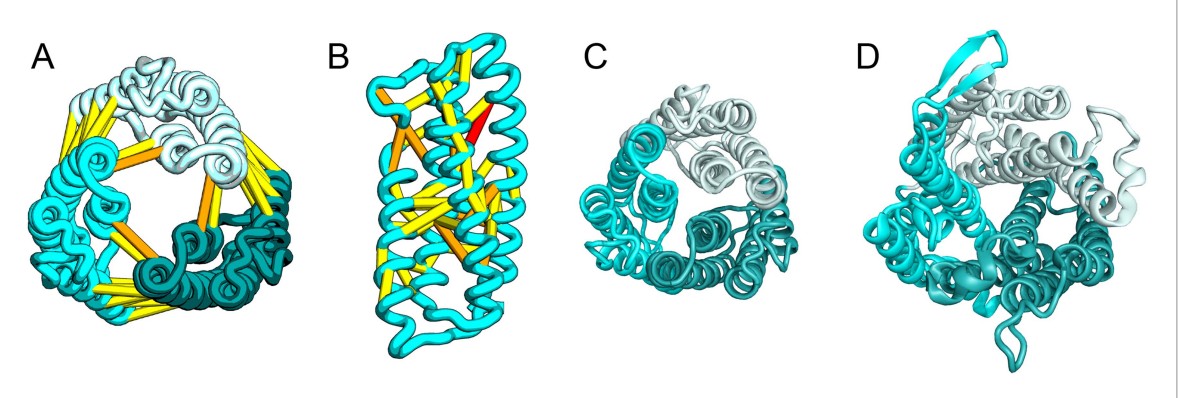

**Figure 15**. *Escherichia coli* protein YgdD. Our model of the *E. coli* protein YgdD trimer (**C**) is based on predicted contacts satisfied within the monomer (**B**) and between monomers in the homo-trimer (**A**). Structural similarity to heme copper oxidase (**D**) along with a weak HHpred sequence match over part of the protein suggests that YgdD is evolutionarily related to heme–copper oxidases.

of contact prediction using the residue–residue coupling values obtained from the model-fitting procedure is dependent on the number of sequences per length and the relative score (*Kamisetty et al., 2013*). To account for these dependencies, we constructed a model (*Figure 16*) that estimates the probability of being in contact using a pdb30 data set from PISCES (resolution limited to 2.5A or better, from 04 January, 2014; [*Wang and Dunbrack, 2003*]), with length of at least 100 residues. MSAs were generated for each of the 10,358 pdb chains using HHblits (-n 8 -e 1E-20 -maxfilt ∞ -neffmax 20 -nodiff -realign_max ∞), and HHfilter (-id 90 -cov 75) in the HHsuite (*Remmert et al., 2012*). The 3392 pdb chains with more than 10 sequences per length were subsampled to create MSAs with varying number of sequences, which were used to estimate probability of contact (*Figure 16A*). CCMPRED v0.1, a parallel implementation of GREMLIN (*Seemayer et al., 2014*), was used for the subsampled alignments. For CCMPRED, the default maximum number of iterations was modified to 100 to ensure convergence. The remaining 7047 pdb chains with less than 10L sequences were saved as a test set (*Figure 16B*, *Figure 17*). The top 3L/2 scores of residue pairs with sequence separation 3 or greater were normalized by rescaling the range so that the minimal value is 0.5 and average value is 1.0. Contact prediction accuracy was found to be a simple function of the residue–residue normalized coupling value, the number of sequences, the length (*Figure 16—figure supplement 1*), and the sequence separation as shown in *Figure 16*. The sigmoidal fit to these observed frequencies was used to estimate the probability of each contact being formed in the native structure.

To evaluate the significance of a match between predicted contacts and a model, we determined the expected total GREMLIN score over all contacts with sequence separation of 6 or greater using P (contact). To evaluate the fit of a particular model to a predicted contact set, we take the ratio of the actual total GREMLIN score of the model to the expected total score computed as above; we refer to this ratio of observed and expected contact scores as 'Rc' throughout the text. As shown in *Figure 17*, Rc ranges from 0.7 to 1.2 for native proteins, and from 0 to 0.3 when contact maps and structures are randomly paired. The Rc was evaluated over the shortest overlap of the two lengths (contact map length vs pdb length). For homo-oligomer complexes, the Rc score includes all chains across all bio units.

## Co-evolution restraints and Rosetta energy function

Residue-pair-specific distance restraints for use in the Rosetta structure prediction calculations were generated based on the normalized GREMLIN scores. Distance restraints were implemented as sigmoidal functions of the form: $restraint(d) = weight(1 + \exp(-slope(d - cutoff)) + intercept)$, where d is the distance between the constrained $C\beta$ atoms ($C\alpha$ in the case of glycine), the distance cutoffs and slopes are amino acid pair specific (SI Table 3 in *Kamisetty et al., 2013*), and the weight is the normalized Gremlin score multiplied by three to give the contact restraints roughly the same total dynamic range as the Rosetta energy. These distance restraints supplement the Rosetta energy

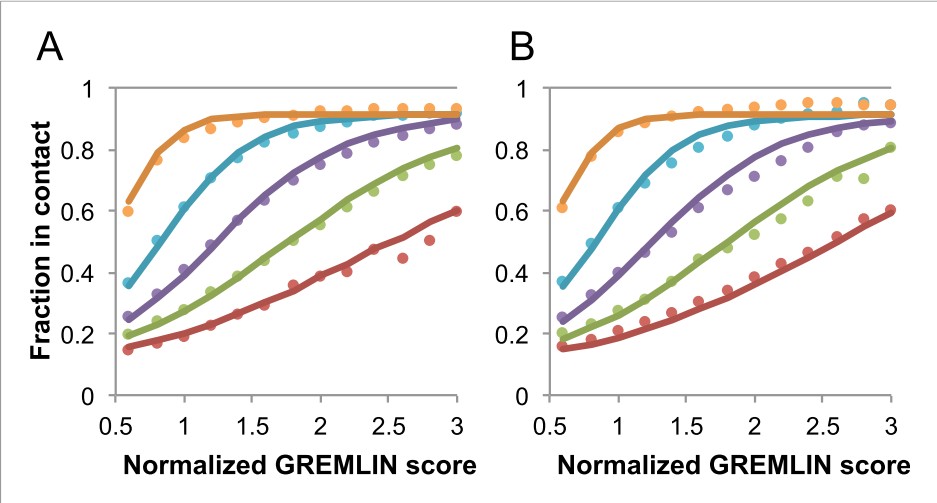

**Figure 16**. Dependence of the accuracy of predicted contacts on the normalized GREMLIN score (sco), the effective number of sequences (seq), the length (len), and the sequence separation (sep). Contacts are defined based on amino acid specific Cβ-Cβ distance cutoffs as described in SI Table 3 in *Kamisetty et al. (2013)*. (**A**) Observed vs predicted accuracies over a large data set of proteins of known structure with deep alignments (*Supplementary file 3*), sub sampled to different extents (seq/√(len) = 4 (red), 8 (green), 15 (purple), 32 (cyan), and 96 (orange)). Circles represent observed contact prediction accuracies, solid lines, a fit to a sigmoid function of the normalized coupling value, the number of sequences, the length, and the sequence separation (see *Figure 16—figure supplement 1* and Figure 16—figure supplement 2). (**B**) Observed vs predicted accuracies in an independent data set of variable length alignments for 7047 pdb chains (*Supplementary file 3*), using maximum number of sequences obtained with HHblits as opposed to subsampling a large alignment. Circles again represent observed contact prediction accuracies; solid lines, the predicted accuracy using the model obtained by fitting to the data in (**A**). The contact prediction accuracy is correctly modeled for the independent data set, justifying its use on the unknown cases described in this article. The Equation use to calculate P(contact|sco,seq,len,sep) is

$$P(\text{contact}|\text{sco}, \text{seq}, \text{len}, \text{sep}) \approx \frac{0.89(1 - P(\text{contact}|\text{sep}))}{1 + \exp\left(-0.58\left(\frac{\text{seq}}{\sqrt{\text{len}}}\right)^{0.50}\left(\text{sco} - 5.46\left(\frac{\text{seq}}{\sqrt{\text{len}}}\right)^{-0.53}\right)\right)} + P(\text{contact}|\text{sep}).$$

The following figure supplement is available for figure 16:

**Figure supplement 1**. Contact prediction accuracy is better correlated with (#sequences/sqrt(length)) than with (#sequences/length).

---

function; the combination ensures the sampling of physically realistic structures consistent with the contact predictions. For TM proteins, the Rosetta energy function was modified to reflect the exposure of non-polar residues in the membrane-spanning regions: the lazaridis-karplus solvation energy term weight was set to zero (fa_sol = 0.00), and to compensate for the short range repulsion implicit in the solvation model, the Lennard-Jones repulsive and attractive terms were given equal weights. We found this simple approach was equally effective and considerably less computationally intensive than the RosettaMembrane approach, which requires estimating the TM region for energy evaluation.

## Model generation

The Rosetta ab initio protocol (*Simons et al., 1999*; *Rohl et al., 2004*) was used to generate 10,000 independent models guided by the covariance-derived restraints. For the benchmark set, fragments database from 2011 was used; for aquaporin, the fragments were filtered to remove any homologs with e-value < 1. After the generation of fragments for aquaporin, we examined the PDB files that contributed the most to the fragment set and verified that they did not contain aquaporin-like structures. The models generated by Rosetta ab initio were refined with an iterative version of the RosettaCM (*Song et al., 2013*) hybridization protocol used to refine models generated with contact information in CASP10

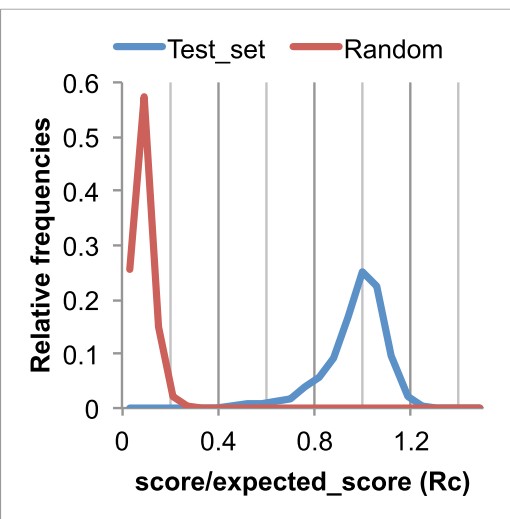

**Figure 17**. The Rc metric used to assess fit of predicted contacts to a model. The expected total GREMLIN score if the structure was native was estimated by summing sco*P(contact|sco, seq, len, sep) over all contacts with sep ≥6. To evaluate the fit of a particular model to a predicted contact set, we take the ratio of the actual total GREMLIN score of the model to the expected total score computed as above; we refer to this ratio of observed and expected contact scores as 'Rc'. Blue line: the distribution of Rc in native structures with 4L–10L sequences; Red line: distribution of Rc after randomly reassigning contact predictions to structures. Rc values less than 0.7 are very infrequently observed for native structures; we use this value as a cutoff to evaluate the fit of a predicted contact set to a model.

(*Kim et al., 2014*). In each iteration, 20 models are produced by recombination and minimization. In addition to the recombination of secondary structure chunks in the input models, fragment insertion was allowed in all positions. Iterations were continued until the procedure converged. For a 200-residue protein, the average runtime to produce a single model is about 30 min for RosettaAB and about 20 min for RosettaCM.

If in the initial Rosetta ab initio calculations, the top 10 models selected by restraint score converged (average pairwise TMscore [*Zhang and Skolnick, 2004*] > 0.8), the top five models were input directly into the iterative RosettaCM hybridization protocol. If models converged over substructures (average pairwise TMscore between 0.5 and 0.8), the top 10 models were first expanded by recombination to a population of 1000 structures, and the top five models were input into the RosettaCM hybridization protocol. If the Rosetta ab initio calculations did not converge (average pairwise TMscore <0.5), we carried out an additional 10,000 Rosetta ab initio trajectories; if the top models did not converge, we considered the structure of the protein not accurately predictable using our approach. 15 of the 121 families were eliminated at this stage. We also eliminated families for which the models generated by the hybridization protocol did not satisfy the predicted contacts; 37 additional families were eliminated at this step.

Proteins over 400 amino acids for which there was little convergence of the lowest energy generated models were parsed into multiple domains (<200 residues) guided by the predicted contact information keeping overlaps of at least 50 residues between each domain, and Rosetta ab initio was used to generate models for each domain separately. If the overlapping regions in each domain converged during modeling, these were used to assemble the full model, otherwise the domains were trimmed to converged residues and docked using RosettaDock (*Chaudhury et al., 2011*).

If models converged overall in the Rosetta ab initio calculations but specific sets of contact restraints were consistently violated, we explored the possibility that the violations correspond to interactions between monomers in a homo-oligomer. To test for oligomeric contacts, docking was performed between two copies of the model using RosettaDock guided by the co-evolution-derived constraints.

## Elimination of non-converging and unconstrained regions

We developed a simple measure of convergence and contact violation after the hybridization protocol to trim regions with higher chance of being in error. The top five percent of the models were selected based on the sum of the Rosetta all atom energy and the contact restraint score and superimposed using THESEUS v3.1 (*Theobald and Wuttke, 2006*). The mean square deviation of the Cα coordinates of each residue was computed, and after smoothing with a Gaussian spanning three residues before and after the central residue, residues with MSD > 2 Å² were trimmed. We also eliminated residues in regions in which there were either very few contact restraints, or the majority of the restraints were violated.

For the benchmark set, the model closest to the average of the lowest energy 5% models was selected, and the RMSD to the native structure was computed over (1) the full length of the protein, (2) the converged and constrained residues, and (3) the residues structurally aligned using TM-align. The latter alignments are longer and more accurate, but selection of the subset of residues requires knowledge of the native structure; this is not the case for (2).

**Table 3**. Comparison of methods on CASP11 targets

| | BAKER* | | Jones-UCL* | | Evfold-web server | |
|---|---|---|---|---|---|---|
| Targets | Cα-RMSD | GDT-TS | Cα-RMSD | GDT_TS | Cα-RMSD | GDT-TS |
| T0806 | 3.6 | 60.4 | 6.8 | 34.3 | 8.2 | 30.0 |
| T0824 | 4.2 | 55.3 | 9.2 | 41.4 | 8.1 | 32.6 |

*Full-length Cα-RMSD and GDT-TS calculation based on the best of five models submitted to CASP11 from BAKER and Jones-UCL groups. For Evfold, the values for best of 50 models generated by the web server are reported, sorted by full-length Cα-RMSD. For the comparison, the alignments used during CASP11 were provided as input to the Evfold-web server, with PLM option selected. For T0824, the minimal number of sequence limit was set to 0 to allow Evfold-web server to run. PLM, pseudo-likelihood.

## Starting structures for homology modeling

Our models provide starting templates to model any member of these families using comparative modeling, which requires relatively little computer time. To evaluate the protein space our 58 models cover, we carried out Jackhmmer search (-E 1E-20 -N 8, [*Eddy, 2009*]) with uniref100 database (*Suzek et al., 2007*) from January, 2015 and eliminated identical sequences and sequences aligned over less than 75% of the protein; the number of remaining sequences for each of the 58 families is listed in *Table 2*.

## Comparison to EvFold server

To compare our method to the EvFold method, we submitted the alignments in our membrane protein benchmark to the EvFold web server. We compared the accuracy of the best of the 50 models generated by the server to our single selected model for each family in *Tables 3, 4*. The Rosetta models are considerably more accurate, but the EvFold server is orders of magnitude faster. We compare to the server results rather than to the results in the previously published Evfold paper as this would be unfair since there were fewer available sequences at the time the paper was written and the contact prediction method (mfDCA) was somewhat less accurate. For the server comparison, we chose to predict contacts using the PLM option as this is very similar to GREMLIN.

**Table 4**. Comparison of methods on transmembrane benchmark set

| | BAKER | | Evfold-web server | |
|---|---|---|---|---|
| Targets | Cα-RMSD | GDT-TS | Cα-RMSD | GDT-TS |
| 4HE8_H | 4.9 | 54.5 | 5.3 | 50.3 |
| 1SOR_A (aquaporin) | 2.7 | 69.7 | 6.1 | 44.5 |
| 4Q2E_A | 5.4 | 45.6 | 12.9 | 21.7 |
| 4HTT_A | 3.9 | 60.6 | 6.4 | 41.8 |
| 4P6V_E | 5.0 | 56.6 | 7.4 | 31.8 |
| 4J72_A | 6.6 | 67.1 | 12.9 | 33.8 |
| 3V5U_A | 3.9 | 58.8 | 4.6 | 47.1 |
| 4PGS_A | 3.5 | 66.3 | 4.6 | 48.1 |
| 4QTN_A | 4.2 | 59.6 | 4.9 | 51.4 |
| 4OD4_A | 3.9 | 55.6 | 4.1 | 53.4 |
| 4O6M_A | 4.1 | 64.0 | 11.2 | 33.0 |

The Cα-RMSD and GDT-TS calculations are over the full sequence. For Evfold web server results, we report the best Cα-RMSD of 50 models returned. For the comparison, the alignments we used were provided as input to the Evfold-web server, and the pseudo-likelihood method was selected.

## Data availability

### External Database

http://gremlin.bakerlab.org/structures/ the external database provides the models generated and links to the GREMLIN web server output. The web server output provides the predicted contacts, overlay of the predicted contacts on the top 10 HHsearch pdb hits, restraints used in modeling and the alignment used for contact prediction. We also provide contact predictions for all protein coding genes with enough sequences for all of the four model organisms used in this paper. The models are also available at Dryad (*Ovchinnikov et al., 2015*).

# Acknowledgements

We thank Per Jr Greisen, Robert Gennis, Ranjani Murali, and Schara Safarian for comments and analysis of CydA/CydB complex, Andrew HJ Wang, and Jason Chou for comments on UppP, Tamir Gonen for comments on YitE, and Mark A Wilson and Rick Lewis for permission to disclose the unpublished structures of YaaA (T0806) and NucB (T0824). We also thank Rosetta@home and Charity engine participants for donating their computer time, Nicole Silvester from ENA and the CASP11 organizers. This work was funded by the NIH/NIGMS and the Welch Foundation.

# Additional information

### Funding

| Funder | Grant reference | Author |
|---|---|---|
| National Institutes of Health (NIH) | R01GM092802 | Hahnbeom Park, David E Kim |
| National Institutes of Health (NIH) | GM094575 | Nick V Grishin |
| Welch Foundation (Robert A. Welch Foundation) | I-1505 | Nick V Grishin |

The funders had no role in study design, data collection and interpretation, or the decision to submit the work for publication.

### Author contributions

SO, HK, NVG, Conception and design, Analysis and interpretation of data, Drafting or revising the article; LK, DB, Conception and design, Acquisition of data, Analysis and interpretation of data, Drafting or revising the article, Contributed unpublished essential data or reagents; HP, Acquisition of data, Analysis and interpretation of data, Drafting or revising the article; YL, JP, Analysis and interpretation of data, Drafting or revising the article; DEK, Acquisition of data, Contributed unpublished essential data or reagents

# Additional files

### Supplementary files

• Supplementary file 1. Detailed table containing all 131 large protein families. Also the complete list of protein coding genes from *E. coli* (ECOLI), *B. subtilis* (BACSU) *Halobacterium salinarum* (HALSA), and *Sulfolobus solfataricus* (SULSO) along with number of sequences are provided.

• Supplementary file 2. Detailed table containing 58 proteins modeled.

• Supplementary file 3. Table of 10,440 PDB chains used in the study and the number of sequences for each.

## Major dataset

The following dataset was generated:

| Author(s) | Year | Dataset title | Dataset ID and/or URL | Database, license, and accessibility information |
|---|---|---|---|---|
| Ovchinnikov S, Kinch L, Park H, Liao Y, Pei J, Kim DE, Kamisetty H, Grishin NV, Baker D | 2015 | Data from: Large scale determination of previously unsolved protein structures using evolutionary information | http://dx.doi.org/10.5061/dryad.987j0 | Available at Dryad Digital Repository under a CC0 Public Domain Dedication. |

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
