## [Decision Letter]

Thank you for submitting your work entitled “Large scale determination of previously unsolved protein structures using evolutionary information” for peer review at *eLife*. Your submission has been favorably evaluated by John Kuriyan (Senior Editor) and 3 reviewers, one of whom is a member of our Board of Reviewing Editors.

The following individuals responsible for the peer review of your submission have agreed to reveal their identity: Yibing Shan (Reviewing editor) and Nir Ben-Tal (peer reviewer).

The reviewers have discussed the reviews with one another and the Reviewing editor has drafted this decision to help you prepare a revised submission.

The manuscript by Ovchinnikov et al. represents a bold attempt in predicting structures of proteins without structural templates. The predictions rely on a physical model (Rosetta conformational search) guided by residue–residue contacts derived from a large number of protein sequences. The latter has been an important source of progress in the field of protein structure prediction in recent years. Using evolutionary data in conjunction with a well-calibrated physical model is a more recent development. This manuscript reports atomic-details structural models of 58 proteins, each represents a large prokaryotic protein family of no 3D structures available. While the accuracy of these models will be examined against the crystal structures when they are available, there are reasons to feel optimistic given the impressive performance of the program in the CASP11 blind test. Importantly, these models are available online and some interesting lessons will be learned regardless of the outcome.

In spite of years of structure determination efforts using X-ray crystallography and other experimental techniques, including the various structural genomics projects, the structure of many important proteins has yet to be determined. To some extent, structure prediction can supplement this. Structures of query proteins can be predicted if suitable templates can be detected. However, this approach is limited by our current ability to assign templates to the queries. This situation is changing now because of two major breakthroughs. First, protein sequence data are becoming available at ever increasing rate. With this data there are now hundreds and thousands of homologues for many proteins. With this, we can think of large protein families, rather than single proteins. The second breakthrough is the emergence of new algorithms for accurate detection of evolutionary couplings between pair of amino acids in large multiple alignment of homologues, and the ability to translate these into distance restraints on the relative locations of the amino acid pair in the 3D protein structure. The premise here is that the detected evolutionary coupling (coevolution, compensatory mutations) results from evolutionary pressure, and is therefore indicative of direct contact.

Recently the Baker's lab introduced its own variant for the detection of coevolution. The improvements over previous implementations of coevolution are (**A**) making the distance restraint depended on the strength of the coevolution signal, (**B**) applying restrains between pairs of amino acids that are close in sequence first and adding restraints between remote residues later in the simulation to avoid trapping in wrong conformations, and (**C**) using the ROSETTA framework.

In the current manuscript, the Baker group joins forces with the Grishin lab in a large-scale application of this tool for predicting the structure of 58 of the most popular protein families in bacteria. This is a well thought of and highly important project, and the manuscript reads really well. The scale is particularly impressive; with so many predicted structures the impact of the work is likely to be high. Modeling was carried out professionally (for the most part, see comments below).

Essential points to address:

1) While the results of CASP11 blind test are important and indicative, they cannot serve as a systematic and direct assessment of the accuracy of Rosetta-GREMLIN. It is necessary to directly compare the accuracy of Rosetta-GREMLIN with other leading methods, particularly EVFOLD. With such comparisons, the extent of progress in terms of prediction accuracy brought by this new method will be more clear and the authors' claims of unprecedented accuracy can be measured in specific metrics (e.g. Ca RMSD or TM score) and calibrated appropriately.

2) The angle and the motivation of this paper is reminiscent of the previous paper by Hopf et al. (Cell, 2012), which reported structure predictions, through exploiting evolutions covariance, for a relative larger number of membrane proteins without experimentally resolved structures. Of course the methodology is different here, but additional discussion of previous work along this line will help place the current work in an historical context. There have been structural predictions for some membrane proteins this paper discussed (e.g. adiponectin receptor and YeiH transporters). Briefly discussing these previous predictions and comparing them with the predictions by Rosetta-GREMLIN is needed.

3) The paragraphs in the Introduction describing the method are too long, too technical, and difficult to read. The Introduction should describe the method at a high level in a language accessible by a general audience and the technical details should be moved to the Methods. One reviewer summed up the new ingredients of this method as (**A**) making the distance restraint depended on the strength of the coevolution signal, (**B**) applying restrains between pairs of amino acids that are close in sequence first and adding restraints between remote residues later in the simulation to avoid trapping in wrong conformations, and (**C**) using the ROSETTA framework. This apt summary could be considered as a framework for a brief description of the method.

4) As one reviewer pointed out, much of the descriptive text concerning the specific structure models is too detailed with not enough motivation. A substantial portion of such text could go in a supplement so that the main discoveries are more visible. Alternatively, the text should be revised to be better connected to the main concern of the paper.

---

## [Author Response]

*Essential points to address*:

*1) While the results of CASP11 blind test are important and indicative, they cannot serve as a systematic and direct assessment of the accuracy of Rosetta-GREMLIN. It is necessary to directly compare the accuracy of Rosetta-GREMLIN with other leading methods, particularly EVFOLD. With such comparisons, the extent of progress in terms of prediction accuracy brought by this new method will be more clear and the authors' claims of unprecedented accuracy can be measured in specific metrics (e.g. Ca RMSD or TM score) and calibrated appropriately*.

We did not argue in the original version of the paper that the Rosetta-GREMLIN method is better than those of Jones and Marks/*Sander*, because this is not the point of the paper. However, we appreciate the question of reviewer II about how much the increased computer time required for Rosetta structure predictions increases model accuracy, and more generally, the need for a more thorough comparison to existing methods, in particular EVFOLD. We have added to the manuscript a table comparing the Rosetta-GREMLIN method to the other methods for both the CASP targets and the membrane protein benchmark. For the CASP targets, we compare to the results of Jones who made submissions for the same targets for CASP. For EVFOLD, we used the excellent EVFOLD web server since EVFOLD did not officially participate in CASP11. For the membrane protein set, we do not compare to the benchmark/results of the previously published Evfold paper, as it would be unfair, given there were far less available sequences then and a less accurate contact prediction method (mfDCA) was used. For the comparison in the paper, the alignments we used were provided as input to the Evfold-webserver, and a contact prediction method very similar to GREMLIN (PLM) was selected. The EVFOLD web server returns 50 models, we computed the CA RMSD and GDT-TS for each of them, and selected the best one. The table compares this “best out of 50” selection to the single Rosetta-GREMLIN model chosen without knowledge of the correct structure as described in the text. We emphasize in the text that while the Rosetta models are more accurate, the EVFOLD models require orders of magnitude less computing time.

*2) The angle and the motivation of this paper is reminiscent of the previous paper by Hopf et al. (Cell, 2012), which reported structure predictions, through exploiting evolutions covariance, for a relative larger number of membrane proteins without experimentally resolved structures. Of course the methodology is different here, but additional discussion of previous work along this line will help place the current work in an historical context. There have been structural predictions for some membrane proteins this paper discussed (e.g. adiponectin receptor and YeiH transporters). Briefly discussing these previous predictions and comparing them with the predictions by Rosetta-GREMLIN is needed*.

We have added a sentence making this excellent point just before the discussion of the models. As described in the submitted manuscript, our adiponectin prediction is similar to the EVfold prediction, while in the other overlap case (EamA) the two models are quite different. We found no match of the YeiH transporter to any of those modelled in [26] nor the Evfold website.

*3) The paragraphs in the Introduction describing the method are too long, too technical, and difficult to read. The Introduction should describe the method at a high level in a language accessible by a general audience and the technical details should be moved to the Methods. One reviewer summed up the new ingredients of this method as (****A****) making the distance restraint depended on the strength of the coevolution signal, (****B****) applying restrains between pairs of amino acids that are close in sequence first and adding restraints between remote residues later in the simulation to avoid trapping in wrong conformations, and (****C****) using the ROSETTA framework. This apt summary could be considered as a framework for a brief description of the method*.

We have considerably simplified the multiple paragraph description of the method for identifying the families modeled in the paper and moved much of the detail to the Methods and table legend. The description of the method for generating the models is already quite brief (less than one paragraph), and it is difficult to shorten it further.

*4) As one reviewer pointed out, much of the descriptive text concerning the specific structure models is too detailed with not enough motivation. A substantial portion of such text could go in a supplement so that the main discoveries are more visible. Alternatively, the text should be revised to be better connected to the main concern of the paper*.

We have eliminated much of the descriptive text about specific conserved residues that are likely functional in the sections concerning specific structural model. We have emphasized in each individual case the gain from analyzing the coevolution data in the context of the three dimensional Rosetta-GREMLIN models over what could be gleaned from the co-evolution and conservation data alone.